



# One size fits all? - Calibrating an ocean biogeochemistry model for different circulations

Iris Kriest[1], Paul Kähler[1], Wolfgang Koeve[1], Karin Kvale[1], Volkmar Sauerland[1,2], and Andreas Oschlies[1]

[1]GEOMAR Helmholtz-Zentrum für Ozeanforschung Kiel, Düsternbrooker Weg 20, D-24105 Kiel, Germany
[2]Department of Mathematics, Kiel University, Christian-Albrechts-Platz 4, D-24118 Kiel, Germany

**Correspondence:** Iris Kriest (ikriest@geomar.de)

**Abstract.**

Global biogeochemical ocean models are often tuned to match the observed distributions and fluxes of inorganic and organic quantities. This tuning is typically carried out "by hand". However, this rather subjective approach might not yield the best fit to observations, is closely linked to the circulation employed, and thus influenced by its specific features and even its faults. We here investigate the effect of model tuning, via objective optimisation, of one biogeochemical model of intermediate complexity when simulated in five different offline circulations. For each circulation, three of six model parameters have adjusted to characteristic features of the respective circulation. The values of these three parameters – namely, the oxygen utilisation of remineralisation, the particle flux parameter and potential nitrogen fixation rate — correlate significantly with deep mixing and ideal age of NADW and the outcrop area of AAIW and SAMW in the Southern Ocean. The clear relationship between these parameters and circulation characteristics, which can be easily diagnosed from global models, can provide guidance when tuning global biogeochemistry within any new circulation model. The results from 20 global cross-validation experiments show that parameter sets optimised for a specific circulation can be transferred between similar circulations without losing too much of the model's fit to observed quantities. When compared to model intercomparisons of subjectively tuned, global coupled biogeochemistry-circulation models, each with different circulation and/or biogeochemistry, our results show a much lower range of oxygen inventory, OMZ volume and global biogeochemical fluxes. Export production depends to a large extent on the circulation applied, while deep particle flux is mostly determined by the particle flux parameter. Oxygen inventory, OMZ volume, primary production and fixed nitrogen turnover depend more or less equally on both factors, with OMZ volume showing the highest sensitivity, and residual variability. These results show a beneficial effect of optimisation, even when a biogeochemical model is first optimised in a relatively coarse circulation, and then transferred to a different, finer resolution circulation model.

## 1 Introduction

Global models of marine biogeochemistry are applied to prognostic problems, such as the future exchange of $CO_2$ between ocean and atmosphere, the evolution of oxygen minimum zones (OMZs) under a changing climate, or future primary production, which is the ultimate food source for fish. Unfortunately, in steady state, these models vary greatly in their representation





of, for example, ocean oxygen inventory (up to 50% of the current value; Bopp et al., 2013) or primary production (varying by more than 90% of present day global production; Bopp et al., 2013). The largest uncertainty is related to OMZ volume – here Bopp et al. (2013) report a range of $\approx 180\%$ of present day volume, for OMZ volumes defined by maximum concentrations of 50 and 80 mmol $O_2$ m$^{-3}$.

Because these coupled models differ both in their physical and biogeochemical setups, to date the contribution of the different
model components to this large variation is not clear (e.g., Cabre et al., 2015). Studies indicate a strong impact of physics on deep oxygen levels, leading to a divergence of up to 150 mmol $O_2$ m$^{-3}$ (Najjar et al., 2007; Seferian et al., 2013). On the other hand, Kriest et al. (2012) and Kriest and Oschlies (2015) showed that the impact of biogeochemical model structure and parameters on deep oxygen profiles can be equally large; also, in the latter study OMZ volume varied among different biogeochemical model setups up to three times the observed value (for an OMZ criterion of 8 mmol $O_2$ m$^{-3}$). Thus, so far
neither oxygen content nor OMZ volume seem well constrained, possibly because of both circulation and biogeochemistry.

In practice, biogeochemical models are often tuned to the corresponding circulation, in order to make the model results (nutrients, oxygen, organic components) to agree better with observations (e.g., Schwinger et al., 2016). Usually this model calibration is carried out "by hand", i.e. by subjectively tuning some biogeochemical model parameters until the models show a "good" fit to the observed tracer fields. The criterion for "good" is not absolute - it may consist of a sufficient visual match, good
indices of some core statistics, (e.g., in Taylor plots; Taylor, 2001), the root-mean-squared error of model results vs. observed quantities, or non-parametric methods such as the Bhattacharya-distance (e.g., Ilyina et al., 2013). Ideally these converge, i.e., they result in a well-defined set of parameters, which provide an optimal fit for all metrics.

However, biogeochemical models include a high-dimensional parameter space with respect to the biogeochemical constants, many of which are not well known. Carrying out a sensitivity study helps to explore a model's sensitivity to its constants (e.g.,
Kriest et al., 2012), but this is a time-consuming task, both in terms of work-hours and computational time. The computational demand is amplified by the fact that global biogeochemical ocean models require a long time to equilibrate (many millennia), owing to the sluggish circulation (e.g., Wunsch and Heimbach, 2008; Primeau and Deleersnijder, 2009), and because biogeochemical processes act in concert with it (e.g., Kriest and Oschlies, 2015). Short spinup times, on the other hand, will produce model results that still depend on initial conditions, and can hamper a thorough assessment of model skill. Because of these
difficulties, there is no common recipe for model spinup (and calibration). This complicates model inter-comparison (Seferian et al., 2016).

Recently, tools have become available to speed up model equilibration, either by efficient offline methods (Khatiwala, 2007), or by root-finding algorithms that solve for the model's steady state (Li and Primeau, 2008; Khatiwala, 2008). Using these tools, automatic calibration of global biogeochemical ocean models becomes more feasible (e.g., DeVries et al., 2014; Holzer et al.,
2014; Letscher et al., 2015; Kriest et al., 2017; Kriest, 2017). However, these approaches have so far mostly been applied to biogeochemical models of low complexity, or to circulation models of rather coarse resolution. As physical processes and resolution can play a large role for the representation of biogeochemical tracer distributions (e.g., Najjar et al., 2007; Duteil et al., 2014), it would be desirable to apply optimisation directly to the more highly resolved models applied in prognostic simulations. Yet, to date this approach seems to be prohibitive, due to the large computational demand mentioned above.





Therefore, we currently must accept a trade-off between finely resolved representation of physical transport processes, and well-tested and objectively optimised biogeochemical models. The calibration of model biogeochemistry in one computationally cheap circulation may elucidate the model's behaviour in its (biogeochemical) parameter space, and indicate a best set of parameters consistent with observed tracer fields. If the mean transport simulated by models were independent from model resolution, one could then transfer these parameters into the more expensive, high-resolution model. However, like

biogeochemical models, physical parameterisations also reflect an idealised system, which can introduce errors in small- and large-scale patterns and processes. Biogeochemical model calibration is affected by these physical errors - the resulting optimal parameters can thus strongly depend on the circulation applied (Löptien and Dietze, 2019). So far it is not clear how model dynamics and performance will change, once these calibrated parameters are transferred to a different circulation, that resolves physical processes in more detail.

To investigate the mutual effects of circulation, biogeochemical model parameters, and model performance we have tested the effect of five different circulations on the objectively optimised parameters of a biogeochemical model. The ocean models differ in resolution as well as physical forcing and dynamics. Biogeochemical model calibration against nutrients and oxygen was carried out using a quasi-evolutionary algorithm, which carries out a dense scan of a six-dimensional, biogeochemical parameter space. Differences in optimal parameters are discussed before the background of large-scale physical properties.

In portability experiments we examine model performance when parameters optimal for one circulation are transferred into another circulation. We finally quantify the effects of changes in parameter sets vs. those of circulation on global quantities such as oxygen inventory, OMZ volume, and global biogeochemical fluxes.

## 2   Models, experiments, and optimisations

### 2.1   Circulation and physical transport

All model simulations and optimisations apply the Transport Matrix Method (TMM; Khatiwala, 2007, 2018) for tracer transport, with monthly mean transport matrices (TMs), wind speed, temperature and salinity (for air-sea gas exchange). One set of TMs and forcing have been derived from a 2.8° global configuration of the MIT ocean model with 15 vertical levels (Marshall et al., 1997). Using this rather coarse spatial grid and a time step length of 1/2 day for tracer transport, a model setup with seven tracers can be integrated for 3000 years in $\approx$ 1-1.5 hours on 4 nodes of Intel Xeon Ivybridge at the North-German

Supercomputing Alliance (www.hlrn.de). This circulation is hereafter referred to as MIT28. For the second physical model configuration (hereafter referred to as ECCO) we apply TMs derived from a circulation of the Estimating the Circulation and Climate of the Ocean (ECCO) project, which provides circulation fields that yield a best fit to hydrographic and remote sensing observations over the 10-year period 1992 through 2001 with a horizontal resolution of $1° \times 1°$ and 23 levels in the vertical (Stammer et al., 2004). A full spinup (3000 years) of the coupled model requires about 9 hours on 16 nodes of Intel Xeon

Ivybridge.

Finally, three sets of transport matrices have been derived from version 2.9 of the University of Victoria Earth System Climate Model (UVic ESCM; hereafter called "UVic"; Weaver et al., 2001), a coarse-resolution ($1.8° \times 3.6° \times 19$ vertical layers) ocean-





atmosphere-biosphere-cryosphere-geosphere model. TM extraction was carried out as described by Kvale et al. (2017). One set of TMs is identical to that described in Kvale et al. (2017), and includes tidal mixing and a high-mixing scheme in the Southern

Ocean, as well as an increased low latitude isopycnal diffusivity (configuration "UHigh"). This configuration utilizes a vertical diffusion coefficient of 0.43 cm$^2$ s$^{-1}$ to stabilise meridional overturning in its linear, 3$^{rd}$ order upwind-biased advection scheme (UW3, Holland et al., 1998; Griffies et al., 2008). This vertical diffusion coefficient is more than double the "standard" value of 0.15 cm$^2$ s$^{-1}$, typically used with the UVic ESCM configured with the default 1$^{st}$ order flux corrected transport (FCT, Weaver and Eby, 1997) advection scheme. Reasons for the change in advection scheme for the application of the TMM to UVic are

given in Kvale et al. (2017), but it is important to note here that the UW3 configuration has not benefitted from the more than two decades of careful parameter adjustments that users of the FCT configuration appreciate. The annual maximum global meridional overturning strength in the UHigh configuration is 18.5 Sv, but other physical features of the circulation have not been previously assessed in detail. Two further sets of UVic ESCM TMs do not include regional adjustments to mixing. They have been tuned to a maximum annual average global overturning circulation of either 20 Sv (named U20) or 17.5Sv (named

U17.5). This tuning was achieved by adjusting the vertical diffusion coefficient (0.409 cm$^2$ s$^{-1}$ in U17.5, and 0.4179 cm$^2$ s$^{-1}$ in U20). The physical circulation parameterisation is otherwise identical to UHigh; utilising UW3 advection and the same tidal mixing scheme. Differences arising in the calibrations between UVic ESCM TMs therefore reflect both differences in the application of regional mixing "corrections" (UHigh versus U17.5 and U20), as well as in global overturning strengths and secondary effects from changed values of the vertical diffusion coefficient. We note that none of the configurations accurately

represent the circulation of the most commonly used UVic ESCM FCT configuration (e.g., Weaver et al., 2001; Schmittner et al., 2005; Somes et al., 2013).

## 2.2 Properties of circulation models

The five circulations differ in many aspects. First, being supported by observational data, ECCO's spatial salinity and density distribution agrees very well with observations, while the MIT28 and UVic circulations show, for example, a too shallow depth

of the $\sigma = 27.5$-isopycnal in the Atlantic Ocean, and too saline waters in the deep northern North Atlantic (Figure 1 and S1). In addition, the three UVic circulations all suffer from a too weak formation and northward propagation of Antarctic Intermediate Waters (AAIW), as identified from water of low salinity at $\approx 1000$m depth in the southern hemisphere. MIT28, U20 and U17.5 also show a too large outcrop area of dense waters ($\sigma_\theta \geq 27.5$) in the Southern Ocean, which does not agree with the observed pattern. Here ECCO and UHigh better match observations.

Striking differences also occur with respect to the annual maximum mixed-layer depth, as derived from a potential density difference to the surface of $\Delta\sigma_\theta \geq 0.03$ (Figure 2), calculated from the models' monthly mean temperature and salinities. Obviously, MIT28 shows a too large area of deep mixing around $60°$S in the Southern Ocean, which does not agree with mixed-layer depths derived from observed temperature and salinity. On the other hand, only this circulation exhibits deep mixing in the Labrador Sea, which is in agreement with observations. All configurations of UVic exhibit a too large area of

deep mixing in the Southern Ocean, while ECCO tends to underestimate mixing in this area.



Differences between circulations are also reflected in the global distributions of ages diagnosed from the models. MIT28, U20 and U17.5 show very old ($> 1400$ years) waters in the deep northern North Pacific (Figure 3). Here, ECCO and UHigh contain much younger waters, mostly below 1400 years. In MIT28 the age increases rapidly with depth in the Southern Ocean (up to more than 800 years), despite its large area of deep mixing. Especially ECCO, but also UVic exhibit much younger

waters below 2000 m in this region. Finally, the U20 and U17.5 configurations result in too old deep waters in the northern North Atlantic (Khatiwala et al., 2012, their Fig. 4). In general, ECCO and UHigh agree much better with mean age constrained with tracer observations (Khatiwala et al., 2012).

To summarise, our applied offline circulations for MIT28 and ECCO differ strongly with respect to resolution and many global physical properties, with the UVic configurations in between these two. As a data-constrained circulation, ECCO shows

the best overall agreement with observations of all five circulations.

## 2.3 Derived indicators of circulation

Based on this first analysis of circulations, we derived three quantities possibly influencing the selection of biogeochemical parameters during optimisation.

**Area of deep mixing:** Deep mixing in the North Atlantic supplies oxygen to the ocean (Khatiwala et al., 2012). In the

Southern Ocean mode and intermediate waters acquire their biogeochemical signatures north of the Antarctic Circumpolar Current (ACC), before being subducted into the interior ocean. These waters then ventilate the thermocline of the subtropics in the southern hemisphere (Sallee et al., 2013). However, as also shown in other studies (for example, Sallee et al., 2013, who found a large variability in Southern Ocean mixed-layer depths simulated by 21 ocean circulation models) the circulations applied in our study differ strongly in the extent and location of deep mixing in this region. To account for the potential effects

of this variability on optimal parameter choice, we evaluated the area of annual maximum deep mixing in the two regions. Mixed-layer depth was defined by a density difference of $\Delta\sigma_\theta \geq 0.03$ (in line with de Boyer Montégut et al., 2004; Dong et al., 2008; Sallee et al., 2013), calculated from monthly mean potential temperature and salinity. For the Southern Ocean (south of $40°$S) and the North Atlantic (north of $40°$N) we then calculated the area, where the annual maximum mixed-layer exceeds either 200 or 400 m (the range of mixed-layer depths simulated and observed in the Southern Ocean; Sallee et al., 2013).

Altogether, we thus obtain four different indicators for ocean ventilation through deep ocean mixing.

**Outcrop area of mode and intermediate water masses:** On centennial timescales the Antarctic Intermediate Water (AAIW) and Subantarctic Mode Water (SAMW) formed in the Southern Ocean determine nutrient concentrations in subtropical areas. Their nitrate deficit and isotopic composition carries signatures of denitrification and nitrogen fixation (Rafter et al., 2013; Tuerena et al., 2015). Given that the models differ so strongly with respect to the surface density in the Southern

Ocean (Figure 1), we evaluated the outcrop area of waters defined by a density of $26.5 \leq \sigma_\theta < 27.5$ and $27.5 \leq \sigma_\theta$ in both the Southern Ocean and the northern North Atlantic (defined as above). In the Southern Ocean the first criterion approximately reflect SAMW and AAIW combined, and the second criterion Circumpolar Deep Water (CDW) (similar to the definitions used by Palter et al., 2010; Iudicone et al., 2011; Rafter et al., 2012, 2013). North Atlantic waters defined by densities of





$26.5 \leq \sigma_\theta < 27.5$ and $\sigma_\theta \geq 27.5$ coincide mainly with the region between 40°N-60°N and the Greenland Sea, respectively

(Figure 1).

**Age of water masses:** We use the concept of water mass (or ventilation) age as a diagnostic for the combined effects of ocean circulation, mixing and ventilation on the time elapsed since a water parcel has been isolated from the atmosphere. We distinguish between average ideal age in three different water masses by applying the criteria of Matsumoto et al. (2004). According to their water mass definitions, North Atlantic Deep Water (NADW) comprises all waters in the North Atlantic

between 0° and 60°N and 1500-2500 m depth. North Pacific Deep Water (NPDW) is defined for a region between 0° and 60°N and 1500-5000 m depth. Finally, Circumpolar Deep Water (CDW) consists of all waters south of 45°S, for a depth between 1500-5000 m (see also Figure 3). We note that, using these region definitions, the average age of NADW is influenced by waters of the Eastern Tropical Atlantic (ETA), which are quite old in ECCO circulation ($> 400$y), and young in UHigh (Figure 3, left panels). One reason for this could be different rates of overturning, which is between 13 to 14 Sv in ECCO (together with

a weak western boundary current; Wunsch and Heimbach, 2006), while the UVic configurations are characterised by higher overturning around 17.5 to 20 Sv. Finally, because the eastern tropical Pacific (ETP) is the main region of fixed nitrogen loss in the models (Kriest and Oschlies, 2015), but the water age in this region varies strongly among the circulations applied in our study (Figure S2), we also calculated average age in the Eastern Equatorial Pacific (ETP) between $\pm 20°$ latitude, east of 160°W, and within 150-500 m depth as a fourth potential indicator.

## 2.4 The biogeochemical model

For all model simulations we apply the Model of Oceanic Pelagic Stoichiometry ("MOPS"), which simulates the biogeochemical cycling among phosphate, phytoplankton, zooplankton, dissolved organic phosphorus (DOP) and detritus. The model is described in detail in Kriest and Oschlies (2015), and we here only give a brief overview on model structure, with focus on parameters (and processes) affected by optimisation (see below). All components are calculated in units of mmol P m$^{-3}$. We

assume a constant nitrogen-to-phosphorus ratio of organic matter of 16 [mol N:mol P]. The oxygen demand of aerobic remineralisation is given by $R_{-\mathrm{O2:P}}$ [mol O$_2$:mol P], following the stoichiometry by Paulmier et al. (2009). In the model oxygen-dependent aerobic remineralisation of organic matter follows a saturation curve with half-saturation constant $K_{\mathrm{O2}}$ [mmol m$^{-3}$]. With declining oxygen, denitrification takes over as long as nitrate is available above a defined threshold $DIN_{\min}$ [mmol m$^{-3}$]. Suboxic remineralisation (denitrification) also follows a saturation curve for the oxidant nitrate, defined by the half-saturation

constant for nitrate, $K_{\mathrm{DIN}}$ [mmol m$^{-3}$]. The model assumes immediate coupling of the different processes involved in nitrate reduction to dinitrogen, following the stoichiometry derived by Paulmier et al. (2009). Loss of fixed nitrogen (through denitrification) is balanced by a temperature-dependent parameterisation of nitrogen fixation, which relaxes the nitrate-to-phosphate ratio to $d$ with a maximum rate $\mu_{\mathrm{NFix}}$ [$\mu$mol m$^{-3}$ d$^{-1}$]. Detritus sinks with a vertically increasing sinking speed: $w = a\,z$ [m d$^{-1}$]. With a constant degradation rate $r = 0.05$ d$^{-1}$, in equilibrium this is equivalent to a depth-dependent particle

flux curve, corresponding to a power law: $F(z) = (z/z_0)^{-b}$, with $b = r/a$ (see Kriest and Oschlies, 2008). Depending on the rain rate to the sea floor, a fraction of detritus deposited at the bottom of the deepest vertical box is buried in some hypotetical sediment. Non-buried detritus is resuspended into the deepest box of the water column, where it is treated as regular detritus.



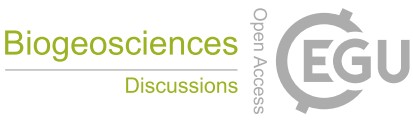

The global annual burial of organic phosphorus and nitrogen is resupplied in the next year via river runoff (when simulated
with MIT28 or ECCO TMs) or at the ocean surface (TMs derived from UVic). We note that in uncalibrated models (e.g., Kriest
and Oschlies, 2015) this creates differences of several percent in the global distribution and inventory of nutrients and oxygen,
comparable in magnitude to those caused by the numerical sinking scheme of detritus (Kriest and Oschlies, 2011). The effect
of this process is subject to further research.

### 2.5  Optimisation algorithm

Optimisation of $n = 6$ biogeochemical model parameters (see subsection 2.7 for choice of parameters to be optimised) is
carried out using an Estimation of Distribution Algorithm, namely the Covariance Matrix Adaption Evolution Strategy (CMA-
ES;  Hansen and Ostermeier, 2001; Hansen, 2006). The application of this algorithm to the coupled biogeochemistry-TMM
framework has been presented in detail in Kriest et al. (2017), and we here only give a brief overview: In each iteration
("generation") the algorithm defines a population of 10 individuals (biogeochemical parameter vectors of length $n$), sampled
from a multi-variate normal-distribution in $\mathbb{R}^n$. Following the simulation of these 10 model setups to near-steady state (3000
years), the misfit (cost) function (presented in subsection 2.6) is evaluated, and information of the current as well as previous
generations is used to update the probability distribution in $\mathbb{R}^n$ such that the likelihood to sample solutions resulting in a
good fit to observations increases. Therefore, the population (the number of model simulations per generation) in CMA-ES
is smaller, and of lower computational demand than in classical evolutionary algorithms, making the algorithm applicable to
this computationally expensive problem. On the other hand, with its quasi-stochastic sampling CMA-ES can still to a certain
degree perform well with misfit functions characterised by a rough topography (i.e., many local optima; Kriest et al., 2017;
Kriest, 2017).

### 2.6  Misfit (cost) function

As in Kriest et al. (2017) and Kriest (2017) the standard misfit to observations $J$ is defined as the root-mean-square error
(RMSE) between simulated and observed annual mean phosphate, nitrate, and oxygen concentrations (Garcia et al., 2006a, b),
mapped onto the respective three-dimensional model geometry. Deviations between model and observations are weighted by
the volume of each individual grid box, $V_i$, expressed as the fraction of total ocean volume, $V_T$. The resulting sum of weighted
deviations is then normalised to the global mean concentration of the respective observed tracer:

$$J_{\mathrm{RMSE}} = \sum_{j=1}^{3} J(j) = \sum_{j=1}^{3} \frac{1}{\overline{o}_j} \sqrt{\sum_{i=1}^{N} (m_{i,j} - o_{i,j})^2 \frac{V_i}{V_T}} \tag{1}$$

$j = 1, 2, 3$ indicates the tracer type (phosphate, nitrate, oxygen) and $i = 1, ..., N$ are the model locations of $N$ model grid
boxes. $\overline{o}_j$ is the global average observed concentration of the respective tracer. $m_{i,j}$ and $o_{i,j}$ are simulated and observed
concentrations, respectively. By weighting each individual misfit with volume, $J_{\mathrm{RMSE}}$ serves as a long time-scale geochemical





estimator in contrast to a misfit function focusing, e.g., on (rather fast) turnover in the surface layer, or resolving the seasonal cycle.

## 2.7 Parameter optimisations and cross-validation experiments

Kriest et al. (2017) applied MOPS with TMs derived from MIT28 to optimise four parameters related to plankton growth and loss terms (in particular: light and nutrient affinity of phytoplankton, as well as zooplankton growth and mortality), together with $b$ and $R_{-O2:P}$. The misfit between simulated annual average nutrients and oxygen after 3000 years and observation was computed as described in Equation 1. The optimal parameters of this first calibration led to a better agreement of simulated biogeochemical fluxes to observations of primary and export production, zooplankton grazing, particle flux at 2000 m, and

organic matter burial at the sea floor (Kriest et al., 2017). In a subsequent optimisation Kriest (2017) kept the optimised plankton parameters fixed and calibrated four parameters related to remineralisation and nitrogen fixation (described above), together with $b$ and $R_{-O2:P}$, against the same data set and cost function. This second optimisation by Kriest (2017) led to a good match to independent estimates of pelagic denitrification, and is hereafter referred to as MIT28*.

Based on calibration MIT28*, we repeated the optimisation by Kriest (2017) against Equation 1 in circulations ECCO,

UHigh, U20 and U17.5 described above. In the following we refer to these four additional optimisations and optimal parameter sets as ECCO*, UHigh*,U20* and U17.5*. To test the portability of optimised parameters to different physical settings, we then transferred the parameter sets of MIT28*, ECCO*, UHigh*,U20* and U17.5* to the other four circulations, again simulating each coupled model for 3000 years. Thus, we present results from 25 different model simulations with five different parameter sets and five different circulations.

## 3 Results

### 3.1 Performance of optimised models

When optimised for different circulations the coupled models show similar values of the misfit function $J^*$ (Table 1). The misfit decreases with more realistic circulation and physics (according to the criteria described in section 2.2), and is lowest for ECCO*. Global mean phosphate profiles are quite similar, and vary less then 10% of the observed concentrations at depths

below 500 m (Figure 4, panel (A)). The low variation of phosphate might be explained by the fixed phosphorus inventory of the model. Only at the surface, where nutrient concentrations become low, relative variation is larger. However, despite optimisation some regional mismatches remain: for example, the model when optimised for the three UVic circulations shows a considerable overestimate of deep (> 3000 m) phosphate in the Atlantic (Figure S3). All optimal models further underestimate phosphate in the mesopelagic of the northern North Pacific.

Vertical nitrate profiles are also quite similar to each other, although the nitrate inventory is allowed to adjust dynamically to the loss of fixed nitrogen during denitrification, and its balance through nitrogen fixation at the sea surface. Because optimisation also attempts to match nitrate observations, the differences between the models are nevertheless quite small, and





result in deviations of 1-2% of observed global mean nitrate (see Table 1). The global pattern of nitrate residuals (Figure S4) generally corresponds to that of phosphate. However, the spatial distribution of fixed nitrogen gain and loss causes variations

in the global distribution of the nitrate deficit in relation to phosphate, as expressed as $N^* = NO_3 - 16 \times PO_4$ (Figure 5).

In agreement with observations, all optimised models show a large nitrate deficit in the Pacific Ocean, manifest in strongly negative $N^*$ (Figure 5). This lack of nitrate is caused by denitrification in the ETP, and is balanced by nitrogen fixation in tropics and subtropics (see Figure S5). In the Atlantic Ocean, where denitrification is largely absent, simulated $N^*$ is far less negative than in other areas, but it is never positive as suggested by the observations. This mismatch can be explained by the

prerequisites of the biogeochemical model applied: In MOPS, nitrogen fixation relaxes nitrate to $16 \times$phosphate with a time constant defined by $\mu_{NFix}$ whenever $N^*$ is negative; otherwise it does not occur. Because of this process parameterisation, $N^*$ is restricted to values $\leq 0$. Finally, the Southern Ocean has moderate values of $N^*$, owing to the mixing of water masses of different origin. Thus, although global average nutrient profiles match observations well, with little differences among optimal models, some regional biases remain, which differ among the models. Also, because of the different processes involved in

nutrient turnover, phosphate and nitrate distributions are not exactly the same, with consequences for the nitrate deficit in different oceanic domains.

Global mean oxygen profiles of optimised models vary more strongly than those of nutrients, up to $\approx 40$ mmol $O_2$ m$^{-3}$ in the deep ocean, and thus more than 20% the observed value (Figure 4). Overall, a finer resolution and a more realistic circulation improve the representation of this tracer, reducing the global oxygen bias, which ranges from 5.3 mmol $O_2$ m$^{-3}$

(U17.5$^*$) to almost zero (ECCO$^*$; Table 1). On a regional scale all models show some common biases: South of 40°S all optimised models overestimate zonal mean oxygen in subsurface waters above $\approx 1500$ m (MIT28$^*$) to $\approx 2000$ m (ECCO$^*$), or even further downward (UVic simulations; Figure S6). In the Pacific Ocean these too high oxygen concentrations propagate northward. Finally, in mesopelagic waters of the northern North Pacific all models overestimate oxygen, especially above 1000 m. Common to all models is further an underestimate of oxygen in subsurface waters (down to $\approx 1000$ m) in the subtropics

and tropics of the southern hemisphere. The Atlantic Ocean is generally characterised by too low oxygen concentrations at greater depths. Here, the five optimal models differ: ECCO$^*$ exhibits too low mesopelagic oxygen in the tropical and subtropical Atlantic. MIT28$^*$ underestimates oxygen particularly in deep waters of the southern hemisphere, and north of 60°N. The optimal UVic configurations are biased low in deep waters of the northern hemisphere. These low oxygen concentrations of the UVic configurations are accompanied by too high phosphate and nitrate in the deep North Atlantic (Figure S4), and indicate

too high remineralisation of organic matter in these depths. Together with circulation these result in a too strong accumulated signal of remineralisation.

Thus, while there are common features among the five optimal models, there are also some striking differences especially in the Atlantic. These differences can be explained with the large impact of the Pacific on the misfit function. Owing to its large volume, optimisation will attempt to minimise especially oxygen misfits in this region, and tune the biogeochemical model

parameters to compensate for potential errors of the respective circulation. These different parameters affect the oxygen distribution in the Atlantic, which does not contribute so much to the global misfit. Different physical properties of the circulations then cause divergent patterns in the oxygen distribution of this region.





## 3.2 Best parameters of optimisations in different circulations differ

As presented and discussed by Kriest (2017), optimisation of MOPS in the circulation of MIT28 reduces the particle flux

length scale by increasing $b$ to 1.39. It further results in a high nitrate threshold for the onset of denitrification ($DIN_{\min} =$ 15.8 mmol m$^{-3}$), a low affinity of denitrification to nitrate ($K_{DIN} = 32$ mmol m$^{-3}$), and a low maximum nitrogen fixation rate ($\mu_{NFix} = 1.19\,\mu$mol m$^{-3}$ d$^{-1}$; Table 1). The optimised oxygen affinity of remineralisation is very high, as indicated by a low value of $K_{O2}$ (the half-saturation constant for oxygen). We note that $K_{O2}$ also regulates the inhibition of denitrification by oxygen; when this parameter becomes very low, denitrification is more strongly inhibited by oxygen. Hence, the optimal model

configuration MIT28$^*$ induces only moderate denitrification, and prevents a decline of the global nitrate inventory through this process (see also Kriest and Oschlies, 2015). The oxygen demand of remineralisation, $R_{-O2:P}$, remains close to the value derived from observations ($170 \pm 10$ mmol O$_2$:mmol P; Anderson and Sarmiento, 1994). Finally, as noted by Kriest (2017), some parameters are only weakly constrained by the misfit function: for example, an almost ten-fold increase in $K_{O2}$ results in a misfit function not larger than 1% of the optimal misfit (see also Table 1). One reason for this low sensitivity of the misfit

to a variation in $K_{O2}$, $K_{DIN}$ or $DIN_{\min}$ is the small volume occupied by suboxic zones, where these parameters can play a role for dissolved inorganic tracer concentrations (Kriest, 2017).

Optimising the same set of parameters in either the three different UVic circulations or the ECCO circulation also results in a high threshold for the onset of denitrification, $DIN_{\min}$ (Table 1). As for MIT28$^*$ the dependencies of oxic and suboxic remineralisation on oxygen or nitrate, expressed through $K_{O2}$ and $K_{DIN}$, may vary largely within the parameter space without

having a large impact on the misfit function.

In contrast, $R_{-O2:P}$ and $b$ are constrained very well by the misfit function, as indicated by the narrow range of parameters that result in a good fit to observations. For example, all solutions of the ECCO$^*$, which result in a misfit within 1% of the optimal fit require a $b$ value between 1.4 and 1.5, and a stoichiometric demand for oxygen between 150 and 154 mol O$_2$: mol P (Table 1). Optimal $R_{-O2:P}$ decreases from 170 mol O$_2$: mol P (MIT28$^*$) over 162 mol O$_2$: mol P (UHigh$^*$) to 151 mol O$_2$: mol P

(ECCO$^*$). The exponent determining the shape of the particle flux curve, $b$, also varies among the five optimisations, between 1.27 (UHigh$^*$) and 1.46 (ECCO$^*$). The range of good (within 1% of the optimal fit) values for $b$ differs between ECCO$^*$ and UHigh$^*$ ($b$ between 1.2-1.3). Also, the range for good values for $R_{-O2:P}$ of ECCO$^*$ does not overlap with that of the other optimisations.

Optimal $\mu_{NFix}$ also varies considerably among the different optimisations, from 1 to 3 $\mu$mol m$^{-3}$ d$^{-1}$. Here, the range

of good parameter values for U20$^*$ (1.0-1.4 $\mu$mol m$^{-3}$ d$^{-1}$) does not overlap with that of ECCO$^*$, UHigh$^*$ and U17.5$^*$ (all between 1.5-3 $\mu$mol m$^{-3}$ d$^{-1}$). Overall, MIT28$^*$ and U20$*$ benefit from a low maximum nitrogen fixation rate around 1 $\mu$mol m$^{-3}$ d$^{-1}$, while the other models require a larger rate between 2-3 $\mu$mol m$^{-3}$ d$^{-1}$.

Therefore, to achieve a good fit to observations different circulations seem to require markedly different parameters for the oxygen utilisation by remineralisation, ($R_{-O2:P}$), the exponent determining the particle flux curve ($b$) and the potential rate of

nitrogen fixation ($\mu_{NFix}$). Other parameters vary little ($DIN_{\min}$), or (as indicated by overlapping ranges of good parameters) the differences among them might not be relevant ($K_{O2}$ and $K_{DIN}$) for a misfit function targeting on the global scale. It is





important to note that the relevant parameters do not seem to be correlated with each other (Figure S7). Apparently, different characteristics of each circulation influence the choice of the optimisation algorithm for the optimal values of $R_{-\mathrm{O2:P}}$, $b$ and $\mu_{\mathrm{NFix}}$.

To investigate the potential dependence of optimal parameters on circulation, we examined the area of dense-water outcrop and deep mixing in two different regions (see section 2.3 for definition). Together with average age of four different regions or water masses, we investigate 12 different diagnostics for each circulation for their influence on optimal biogeochemical parameter estimates. In most cases the optimal parameters are not correlated with physical properties (Table 2). However, the oxygen demand of remineralisation, $R_{-\mathrm{O2:P}}$ is significantly correlated with the area of deep mixing in the northern North
Atlantic for maximum mixed-layer depths of 200 and 400 m ($p < 0.05$). For both criteria $R_{-\mathrm{O2:P}}$ increases with increasing area of deep mixing (Figure 6). In addition, $R_{-\mathrm{O2:P}}$ also correlates with a deep mixing area defined by $> 200$ m in the Southern Ocean, albeit not significantly (Table 2). In other words, more vigorous mixing in areas of deep, intermediate or mode water formation allows for a higher oxygen utilisation by remineralisation. Parameter $b$ describing the particle flux curve correlates significantly with the ideal age of NADW (Table 2), and increases with increasing age of this water mass (Figure 6).
Finally, the maximum potential rate of nitrogen fixation $\mu_{\mathrm{NFix}}$ correlates with the outcrop area of waters with moderate density ($26.5 \leq \sigma_\theta < 27.5$) waters in the Southern Ocean. An increase in the outcrop area of these waters - which correspond roughly to the sum of AAIW and SAMW - results in an increase of optimal maximum nitrogen fixation rate (Figure 6). The age of mesopelagic waters in the ETP seems to play a small role, despite the fact that it is an important area in the global nitrogen budget (Figure S5). Possible reasons for the dependence of these three parameters on physical diagnostics will be discussed in
section 4.1.

### 3.3 Cross-validation experiments: Can we transfer parameters optimal for one circulation to another circulation?

Given that three optimal parameters differ among the model circulations, we investigate model performance and dynamics when these parameter sets are swapped among circulations. Of course, every coupled model performs best (with respect to $J^*$ of Equation 1) when simulated with parameters optimal for the respective circulation, as indicated by the lowest relative
misfit along the main diagonal in panel (A) of Figure 7. When exchanging the biogeochemical parameters optimal for ECCO or MIT28 circulation with parameters optimised in any other circulation, the model performance with respect to respect to $J^*$ deteriorates. MOPS coupled to UVic circulation is more robust with respect to changes in parameters.

Likewise, the global oxygen bias is low for each optimal model configuration, indicated by the low values along the main diagonal of Figure 7, panel (B). Likely, the large impact of oxygen on the misfit function (Kriest et al., 2017) causes the good
representation of the oxygen inventory in the optimised models. The oxygen bias induced by the changes in parameter set and circulation depends on the combination of these two: For example, the low value for $R_{-\mathrm{O2:P}}$ and the high value for $b$ of ECCO* causes a large overestimate of the oxygen inventory in any other circulation (indicated by warm colours in the second column of Figure 7, panel (B)). Vice versa, applying optimal parameter sets from MIT28*, UHigh*, U20* or U17.5* to the ECCO circulation results in a too low oxygen inventory (indicated by cold colours in the second row from the bottom of Figure 7,
panel (B)). When swapping parameter sets among the different configurations of the UVic circulation, the effect is much less





pronounced. Apparently, despite their different overturning and mixing, the UVic circulations are more similar to each other than those of MIT28 and ECCO. We note that these large differences in oxygen inventory arise mainly from deeper ($27.5 \leq \sigma_\theta$) layers, while the oxygen inventories of waters lighter than $\sigma_\theta = 27.5$ are quite similar (Figure S8).

OMZ volume is biased low for the parameter set of ECCO*, and in the MIT28 circulation (Figure 7, panel (C)). This underestimate is likely caused by the very low $R_{-O2:P}$ and high $b$ of ECCO*, or the vigorous mixing in the MIT28 circulation, which both tend to increase subsurface oxygen concentrations. Otherwise, OMZ volume does not seem to be closely related to the parameter set or circulation, likely because this diagnostic is independent of the applied misfit function, and depends on the local circulation pattern (see discussion by Sauerland et al., 2019).

    Thus, because of different physical model properties, the biogeochemical model MOPS, when coupled to the MIT28 and
ECCO circulation requires unique and different sets of parameters for optimal model performance. In the UVic circulations the model is more flexible with regard to parameters; yet, when aiming for independent diagnostics such as OMZ volume, there is no clear relationship between OMZ volume and changes in parameter set or circulation.

### 3.4   Effect of parameters and circulation on phosphate and oxygen concentrations in different water masses

Because of the regional biases of nutrients and oxygen in the North Atlantic, North Pacific and Southern Ocean (see section 3.1),
and because the values of $R_{-O2:P}$ and $b$ selected in the calibration process correlate significantly on water mass properties of the NADW, NPDW and CDW (section 3.2), we here examine more closely how these parameters affect the large scale distribution of phosphate (as a conserved nutrient) and oxygen, which can adjust dynamically at the model's air-sea interface, and is thus a non-conservative tracer.

    Kriest et al. (2012) showed that $b$, the parameter determining the particle flux length scale, has a large influence on the
distribution of phosphate along the "conveyor belt" (i.e., along waters of different age), in agreement with the results obtained by Bacastow and Maier-Reimer (1991) and Kwon and Primeau (2006). We here carry out an analysis similar to that by Kriest et al. (2012) and evaluate average phosphate and oxygen within the NADW, NPDW and CDW, with region definitions as described above for water mass age (subsection 2.3).

    Within each circulation a smaller value of $b$ (corresponding to faster sinking particles) increases phosphate in the NPDW, and
decreases it in the NADW (Figure 8, panel (A)), confirming the pattern found by Kriest et al. (2012). When plotting average phosphate in the NPDW against average phosphate in the CDW there is no such relationship, but the average value in the CDW varies only little (Figure 8, panel (B)), and is near the observed value of 2.26 mmol m$^{-3}$. The spread of phosphate concentrations caused by different parameter sets (same symbol with different colours) is about the same (between 0.1-0.15 mmol m$^{-3}$) as the spread caused by different circulations (different symbols of the same colour). Thus, both biogeochemistry and circula-
tion seem to play an equally large role for the distribution of phosphate between NADW and NPDW. The variation caused by biogeochemical parameters is smaller for CDW, indicating that in this region physical processes play a larger role.

    In contrast to phosphorus, the oxygen inventory is not fixed, but regulated by the interplay of circulation, air-sea gas exchange, and biogeochemical turnover. Because of this we find a pattern that is very different from that of phosphate when examining the distribution of average oxygen in different water masses. Now, within each circulation average oxygen in the





NPDW increases almost linearly with average oxygen in the CDW and NADW (Figure 9), highlighting the role of these waters
for the ventilation of the deep North Pacific. In most circulations, a large value of $R_{-O2:P}$ results in a low oxygen content in all
water masses. All optimised coupled model configurations suggest average oxygen between 220-250 mmol m$^{-3}$ in the NADW.
For the NPDW all optimised models simulate average oxygen concentrations around 150 mmol m$^{-3}$, and thereby overestimate
the observed value of 125 mmol m$^{-3}$ by one fifth. Average oxygen in the CDW of optimal model simulations varies between
$\approx 210 - 240$ mmol m$^{-3}$, encompassing the observed value of 215 mmol m$^{-3}$. Thus, given the quite wide range of potential
parameter values, optimisation improves the global oxygen bias (see Table 1), but some residual regional bias especially in the
NPDW remains.

### 3.5 Effect of parameters and circulation on global oxygen inventory and OMZ volume

A declining trend of global average oxygen with increasing $R_{-O2:P}$ is also reflected in panel (A) of Fig. 10, but circula-
tion also plays a role for the oxygen inventory, with the ECCO circulation showing the lowest values. To have a closer look
at the individual contributions of circulation and biogeochemistry to the overall variability of oxygen inventory and OMZ
volume we have calculated their maximum spread caused by varying only the circulation (keeping the biogeochemical pa-
rameter set constant; ΔCirc) and by varying only the biogeochemical parameters (keeping the circulation constant; ΔPar).
For example, to determine ΔCirc for global average oxygen or OMZ volume (here denoted as $X$), for each parameter set
$i$ simulated with the five different circulations $j = 1...5$ we compute the difference between the maximum and minimum
value $\Delta X_i = \max(X_{i,j=1...5}) - \min(X_{i,j=1...5})$, and then determine the maximum of these differences: ΔCirc$= \max(\Delta X_i)$.
The computation of ΔPar is done analogously. We also compute the maximum across all optimal model configurations
ΔOpt$= \max(X_{i=j}) - \min(X_{i=j})$, and across all 25 experiments presented in this study: ΔAll$= \max(X_{i,j}) - \min(X_{i,j})$.
Using this approach, a value for ΔPar close to ΔAll indicates that the model variability is mainly induced by the biogeochem-
ical parameter set, whereas a relatively large value for ΔCirc indicates a major impact of circulation. Table 3 and Figure 11
show the results of this comparison, and Figure 12 illustrates the variability for each circulation or parameter set, normalised
by the average over all optimal models. Note that the longest horizontal or vertical line in Figure 12 corresponds to ΔPar and
ΔCirc in Table 3, while the width of the grey shaded square corresponds to ΔAll.

Firstly, biogeochemical parameters (ΔPar) as well as circulation (ΔCirc) play an about equally large role for the global
average oxygen, which varies by $\approx 24$ mmol m$^{-3}$ (Table 3 and Fig. 11), or about $\pm 15\%$ of the average optimal value (see
also Figure 12). The variation decreases to less than one fourth of this value if we restrict our analysis to only optimal models
(ΔOpt); as noted above, this strong decrease arises because all optimal models have adjusted $R_{-O2:P}$ to account for different
ventilation in the high latitudes. Considering all 25 experiments, i.e., the accounting for variation induced by both biogeochem-
ical and physical configurations (ΔAll), leads to a variation six times as large as for the optimal configurations (ΔOpt).

The variability is much more pronounced when considering the OMZ volume as defined by two criteria, 50 mmol m$^{-3}$
and 80 mmol m$^{-3}$. Again, both circulation and parameter set play an about equally large role; but the impact of changes in
parameters or circulation varies across the different models (Fig. 12). For example, applying the parameter set of MIT28$^*$ to a
different circulation causes a very strong increase in OMZ volume (vertical black lines in Fig. 12), while model MOPS coupled





to the circulation of MIT28 is quite robust with respect to different parameters (horizontal black lines in Fig. 12). On the other

hand, when coupled to the ECCO circulation the biogeochemical model is quite sensitive to the biogeochemical parameter set (horizontal red lines in Fig. 12), but its optimal parameter set ECCO* has a smaller effect when applied to other circulations (vertical red lines in Fig. 12). This diverging effect of parameters and circulation among the different models eventually causes a large spread of $67.2 \times 10^{15}$ m$^{-3}$ of global OMZ volume across all model experiments ($\Delta$All in Table 3 and Fig. 11). The effects are even more pronounced when considering a criterion of 80 mmol m$^{-3}$ for OMZ definition (Table 3). The OMZ

volume does not show any consistent trend with $b$, $R_{-\mathrm{O2:P}}$, or circulation (Fig. 10, panel (B)), although models with high $b$ and low $R_{-\mathrm{O2:P}}$ tend to result in a smaller OMZ volume. The circulation of MIT28 shows the lowest OMZ volume.

To summarise, oxygen inventory and OMZ volume are almost equally influenced by physics and biogeochemistry. Optimisation reduces the spread induced by either biogeochemistry or physics to about 30% percent for average oxygen, but less for OMZ volume, which varies strongly across all model experiments.

### 3.6 Effect of parameters and circulation on global biogeochemical fluxes

Oxygen and nutrient distributions are influenced by the production of organic matter in the euphotic zone, and its subsequent transport to the ocean interior by physical and biogeochemical processes. In addition, denitrification in combination with nitrogen fixation can affect the global nitrogen inventory, and the spatial distribution of the nitrate deficit (see section 3.1). We finally here investigate how these fluxes are affected by the two parameters $R_{-\mathrm{O2:P}}$ and $b$.

In our model experiments simulated global primary production depends slightly more on circulation ($\Delta$Circ) than on biogeochemical parameters ($\Delta$Par; Table 3, Figures 11 and 12). An increase in $b$ (corresponding to slowly sinking particles) causes primary production to increase (Figure 10, panel (D)), likely because of the higher nutrient retention in the euphotic zone, shallow remineralisation and enhanced entrainment of nutrients into the surface layers. Because the latter process depends on physical dynamics, we also find an influence of the circulation model on global primary production. Further, our optimisations

did not include parameters relevant for plankton dynamics at the surface, which can also explain the comparatively large impact of circulation. The variation across all optimal models of our study ($\Delta$Opt) is much smaller (about one third) than the variation across all model experiments ($\Delta$All).

Circulation also plays a large role for export production (particle flux through 100-130 m, depending on model grid), as it supplies new nutrients to the well-lit upper ocean which will, under steady state conditions, be exported again. Somehow

surprisingly, export production is not strongly determined by $b$ (Fig. 10, panel (E)). This parameter affects directly the sinking of organic matter out of the euphotic zone. On the other hand, it determines the subsurface concentration of nutrients, as a source for upwelling and entrainment of nutrients. A large $b$, corresponding to slow sinking and shallow remineralisation, increases nutrients within and below the euphotic zone, and thus primary production; on the other hand, it prevents fast settling of organic particles out of the euphotic zone. The combined effect explains the relatively small variation caused by biogeochemical

parameters on export production ($\Delta$Par; Table 3 and Figures 11 and 12), which is only about half as much the variation due to circulation ($\Delta$Circ).





Deep particle flux, on the other hand, is almost entirely determined by $b$, and circulation plays a negligible role for this flux (Fig. 10, panel (F)). The large influence of this parameter is also reflected in its range over all model simulations, which is only slightly larger than the range of flux in optimally configured models (Table 3). Therefore, simulated organic matter supply to the deep ocean, and long term storage of nutrients and carbon will, to a large extent, depend on the prescribed particle flux profile.

The loss of fixed nitrogen through pelagic denitrification is tightly related to the extent of OMZs, and thus varies quite strongly among the different experiments, with no clear trend for either $b$, $R_{-\mathrm{O2:P}}$ (Figure 10, panel (C)) or $\mu_{\mathrm{NFix}}$ (no figure). The range of variation due to parameters and circulation is about equally large (about 50% of the average optimal global flux; Table 3 and Fig. 11). Overall, this global flux is affected by both circulation and changes in biogeochemical parameters, which induce changes of about $\pm 30\%$ around the mean optimal flux for each model circulation or parameter set (Fig. 12). Global fixed nitrogen loss of the optimised models varies much less, likely because optimisation adjusts the parameters to match observed nitrate profiles.

To summarise, circulation and biogeochemistry affect global biogeochemical fluxes in different ways. While primary production and fixed nitrogen loss are almost equally influenced by physics and biogeochemistry, export production depends mainly on physics. Deep particle flux, on the other hand, is affected to a large extent by $b$. Optimisation reduces the spread induced by changing either biogeochemistry or physics to about 50% percent for fixed nitrogen loss and for primary production (compare $\Delta$ Opt with $\Delta$ Circ or $\Delta$ Par). In contrast, there is no such reduction in model variability for export production (which is mainly determined by circulation) or deep particle flux (which is mainly determined by $b$).

# 4 Discussion

## 4.1 Why do different circulations require different parameters?

As we have seen in section 3.2, three optimal parameters depend significantly on three unique diagnostics that result from different features of the circulation model. These diagnostics are related to the northern North Atlantic and the Southern Ocean; the ETP seems to play a lesser role.

The strong correlation of $R_{-\mathrm{O2:P}}$ with the area of deep mixing clearly confirms that these two model properties (physics and biogeochemistry) regulate global ocean oxygen distribution and inventory in concert. The larger the area of deep mixing, the more oxygen can – or should – be respired in the model, in order to match observed oxygen concentrations. Despite optimisation, the optimal models show an average oxygen concentration of $\approx 145$ mmol m$^{-3}$ in the NPDW (Figure 9), which is higher than the observed value of 125 mmol m$^{-3}$. Given that the models differ strongly in their physical properties, this residual mismatch of all optimal models especially in the NPDW may point towards a deficiency of the biogeochemical model. For example, the spatially homogeneous, and thus inflexible, particle flux profile may not be adequate to simulate the very dynamic response of ecosystem dynamics and particle size structure to regionally variable mixing and nutrient supply (e.g., Guidi et al., 2015; Marsay et al., 2015). Here, a more flexible model resulting in variable sinking speeds of particles (e.g.,





Gehlen et al., 2006; Niemeyer et al., 2019), or, more generally, spatially flexible remineralisation length scales (e.g., Weber
et al., 2016), might be of advantage.

The parameter determining particle flux, $b$, correlates with the ideal age of NADW (Figure 6). This physical diagnostic
comprises several aspects of circulation: a large area of deep mixing in the northern North Atlantic supplies this region with
"young" waters. At the same time, a strong Atlantic Meridional Overturning circulation (AMOC) and/or confined Deep West-
ern Boundary current (DWBC) can more quickly export the preformed properties to the southern parts of the basin. Depending
on the parameterisation of mixing and other physical processes, biogeochemical tracers are distributed more efficiently in the
west-east direction, or mixed with deeper waters. When these combined properties of a model cause a long residence time of
waters in the NADW, the resulting age of this water mass will be quite high, and vice versa.

The circulations applied in our study vary with regard to several aspects in this region: in contrast to all other models the
circulation of MIT28 has a large area of deep mixing in the north, including the Labrador Sea (see Figure 2). There is also a
quite strong and wide transport of young waters in the western part of the North Atlantic via the DWBC, as indicated by the
southward propagation of relatively young waters between 1500-2500 m in the western part of the basin (Figure 3). At the
same time there is a strong lateral spreading of these young waters from the western part (see also Dutay et al., 2002). All
processes combined lead to relatively young average age of NADW in MIT28. ECCO, in contrast, shows only comparatively
shallow mixing in the northern North Atlantic (Figure 2), little southward transport of these waters in the DWBC (Wunsch and
Heimbach, 2006), and a large extent of older waters in the Eastern Tropical Atlantic (Figure 3). This circulation is characterised
by the oldest average age of NADW.

Our optimisations suggest that models with old NADW adjust to a large $b$, or slow particle sinking (for example, $b =$
1.46 of ECCO*). As ideal age becomes younger, optimal $b$ decreases. Why is this the case? So far, we can not attribute this
exclusively to the area of deep mixing (Table 2), or, for example, to the overturning circulation, which is quite low in ECCO
(between 13-14 Sv; Wunsch and Heimbach, 2006), moderate (17.5 and 18.5 Sv) in U17.5 and UHigh, and high (20 Sv) in U20.
Instead, the average age of NADW, and resulting optimal $b$, likely reflects the combined effects of various model physical and
biogeochemical parameterisations: the adjustment of $b$ to smaller values decreases shallow production and remineralisation
(see Figure 10). It also increases export of phosphorus to deep waters, and finally to the NPDW (see Figure 8). Circulation
models with high physical turnover in the NADW (e.g., UHigh), as indicated by young NADW, can more easily resupply
nutrients to surface waters, and therefore balance the loss due to particle sinking in this region.

As shown in Figure 6, the maximum potential rate of nitrogen fixation $\mu_{\mathrm{NFix}}$ increases with area of surface waters defined
$26.5 \leq \sigma_\Theta < 27.5$ in the Southern Ocean, i.e., waters reflecting the formation and ventilation of AAIW and SAMW. A broad
view of large scale circulation and the spatial separation of fixed nitrogen loss and gain helps to understand the adjustment
of maximum nitrogen fixation rate to physical processes in the Southern Ocean. Denitrification is a very localised process,
occurring mainly in the Eastern Tropical Pacific (ETP) (Figure S5). On the other hand, simulated nitrogen fixation takes place
throughout large parts of the tropical and subtropical regions of the Pacific, Atlantic and Indian Ocean. Even though nitrogen
fixation in the Atlantic accounts only for a fraction of global fixed N gain (see also Marconi et al., 2017, for evidence from
observations), in our models it nevertheless contributes to the stabilisation of the global fixed-nitrogen budget. A very negative





N$^*$, as arises from denitrification in the ETP, has to arrive in the Atlantic for nitrogen fixation to trigger a competitive advantage

of nitrogen fixation. These two regions in the Atlantic and Pacific Ocean are connected through large scale circulation, which transports N$^*$ on centennial to millennial time scales from areas of fixed nitrogen loss to areas of fixed nitrogen gain. When passing the CDW of the Southern Ocean, these waters can act as a "mixer of deep waters with distinct isotopic signatures and nutrient stoichiometry" (Tuerena et al., 2015); the resulting mixed properties provide the source of AAIW and SAMW. The subsequent transport via AAIW and SAMW then can trigger nitrogen fixation, e.g., in the Atlantic, and balance the nitrate

deficit arising mainly in the Pacific Ocean.

As shown in Figure 5, the nitrate deficit N$^*$ differs among the different models. For example, MIT28$^*$ exports water with an N-deficit of $\approx 3$ mmol m$^{-3}$ from the Southern Ocean to the low latitudes (promoting nitrogen fixation). This model adjusts to a low rate of maximum potential nitrogen fixation of 1.19 $\mu$mol m$^{-3}$ d$^{-1}$. On the other hand, UHigh$^*$ simulates SAMW and AAIW that contain a lower N-deficit of $\approx 2$ mmol m$^{-3}$, which – depending on phosphate availability – will result in lower

nitrogen fixation. The optimal high parameter of UHigh$^*$ of almost 3 $\mu$mol m$^{-3}$ d$^{-1}$ can partially compensate for this. The effect of N$^*$ is, however, not consistent across all optimal models: U17.5$^*$ also shows a small nitrate deficit in this region, but has a still relatively low maximum nitrogen fixation rate. Here, other effects might play a role, such as a stronger ventilation and consequently younger waters in the ETP (Figure S2), which induce a smaller OMZ (Table 1), less denitrification in this region (Figure S5), and thus a lower nitrate deficit in this area, that is to be eventually balanced by nitrogen fixation.

In our analysis we have combined outcrop area of two water masses, SAMW and AAIW into one single diagnostic. Separating the impact of the two water masses on this parameter, we find that the correlation of $\mu_{\text{NFix}}$ with SAMW outcrop area (when defined as by $26.5 \leq \sigma_\Theta < 27.0$) is less significant ($r = 0.81$) than that with AAIW ($27.0 \leq \sigma_\Theta < 27.5; r = 0.88$), which is somehow in contrast to the findings by Palter et al. (2010). Their model experiments showed that the largest fraction (between 45 to 68%, depending on model configuration) of water volume at the surface between 30°S and 30°N stems from

SAMW, highlighting the role of this water mass for nutrient supply in the tropics and subtropics. A possible explanation for this difference between our results and the results by Palter et al. (2010) could be the slightly different definition of water masses. Further, in our models waters denser than $\sigma_\Theta = 26.5$ are influenced by the small nitrate deficit of surface waters in the subtropical southern hemisphere (Figure 5), which moderates the signal arising from denitrification in the Pacific.

### 4.2   Can optimisation help to improve model performance?

As shown in section 3.2, each circulation requires its own set of parameters for an optimal fit to dissolved inorganic tracers. Optimisation facilitates the identification of these constants; on the other hand, it requires many model evaluations, so this approach is prohibitive for models of high resolution because of computational constraints. It would be desirable to optimise a biogeochemical model in a computationally cheaper circulation and then transfer the optimal parameters to a different model that includes more physical details, but is computationally more expensive. However, as shown in Sections 3.3, 3.5 and 3.6,

model performance can deteriorate when simulated with non-optimal parameters, and result in a considerable spread of the independent diagnostics. Is there any advantage of calibrating biogeochemical models in these rather coarse scale, simplified





circulations, if the parameters are to be transferred to a different circulation? To answer this question, we here discuss the model variability before the background of earlier model studies and observed estimates.

The mean diagnostic across all 25 model experiments (Mean(All) of Table 3) differs only slightly from the mean across only
the optimal model configurations (Mean(Opt)), and is close to observed quantities (Table 3), in agreement with Kriest et al. (2017) and Kriest (2017), who found that optimisation against global nutrients and oxygen can help to improve global model performance. Further, the overall maximum variation across all 25 experiments (ΔAll of Table 3) is usually less than 50% of that found by Bopp et al. (2013), who examined seven global models of different biogeochemical structure and circulation for their global average oxygen, OMZ volume, primary and export production. This indicates that optimisation can help to
improve model performance and reduce its uncertainty, even if parameters were optimised in a different circulation.

### 4.2.1 Model uncertainty, oxygen inventory and OMZ volume

As shown in section 3.5 changes in circulation and biogeochemical parameters affect model performance with respect to global average oxygen about equally, resulting in an overall variation that is less than 25% of the observed value (ΔAll of Table 3). In contrast, the global OMZ volume shows a large response to variations in circulation and model parameters, and varies by more
than 100% (ΔAll of Table 3) to 200% (Bopp et al., 2013) of the observed volume. To our knowledge, no global model study exists so far that systematically distinguishes between the effects of circulation and biogeochemistry on global OMZ volume; our study suggests that both are equally important. Even the range across optimal models (ΔOpt) is still quite large, which can be explained by the fact that the target of optimisation (the RMSE to nutrients and oxygen, as of Equation 1) is only weakly correlated to the fit to OMZs (Sauerland et al., 2019). Application of a revised misfit function, or multi-objective optimisation
as presented by Sauerland et al. (2019), can help to better constrain the relevant model parameters, and better represent OMZs. Nevertheless, the mean of all models in our study deviates by less than 5% from the observed mean. Obviously, a good representation of nutrients and oxygen can improve the fit to OMZs to some extent.

However, many global circulation models suffer from a deficient representation of physical processes in the tropics and subtropics, for example in their representation of the equatorial undercurrent (Dietze and Loeptien, 2013), or from inadequate
ventilation from the Southern Ocean and North Pacific (see Cabre et al., 2015, and citations therein). The first problem can possibly be cured by a higher resolution, which leads to a more realistic OMZ ventilation by equatorial and off-equatorial undercurrents (Duteil et al., 2014). Parameterisation of intermediate jets (Getzlaff and Dietze, 2013) can also lead to a better agreement of the models. Tuning of biogeochemistry before the background of inadequate physics could compensate for the physical errors, but also result in misleading model parameterisations ('overtuning'), with potential consequences for future
projections (Löptien and Dietze, 2019). Given the yet unexplored structural and parameter sensitivity of models employed in global assessments, and the large error with respect to OMZ volume and expansion (Table 3; Cocco et al., 2013; Bopp et al., 2013; Cabre et al., 2015), a careful analysis of different error sources (physical and biogeochemical) can help to determine the reasons for model divergence. The study presented here can serve as a first step towards this.





### 4.2.2    Model uncertainty and global biogeochemical fluxes

The loss of fixed nitrogen through pelagic denitrification is tightly linked to OMZs, and therefore also almost equally influenced by circulation and biogeochemical parameters. Our average optimal model estimates are in agreement with recent estimates by Eugster and Gruber (2012) and Somes et al. (2013). The variation due to biogeochemical parameters is lower than found by Somes et al. (2013), who varied the nitrate threshold for denitrification from 20 to 32 mmol m$^{-3}$, i.e. a wider range than identified by our objective parameter calibration (see Table 1).

Average global primary production of the models lies well within the range of observed estimates (Carr et al., 2006), and depends slightly more on circulation than on biogeochemical parameters, similar to the results obtained in the sensitivity study by Schmittner et al. (2005). That study included a wide range of sinking and mixing parameterisations, which might explain the larger variation compared to our results. Applying three different circulations to one biogeochemical model, Seferian et al. (2013) found a spread of primary production which comparable to our experiments.

Quite many global model studies analysed the impact of circulation on global export production. Najjar et al. (2007) found an effect of circulation more than ten times larger than in the present study, which can likely be explained by the nutrient-restoring approach applied in their simple model. Using a more complex biogeochemical model, export production in the study by Seferian et al. (2013) varied only by $\approx 3$ Pg C y$-1$, which is closer to the effects of circulation found in our study. The large effects observed by Schmittner et al. (2005) can again likely be ascribed to the wide range of parameterisations tested.

Using a biogeochemical model similar to the one applied by Najjar et al. (2007), Kwon et al. (2009) found an increase in export production of $\approx 5$ Pg C y$^{-1}$ when increasing $b$ from 1.1 to 1.4 (about the range tested in our study). Again, this can possibly be explained by the nutrient restoring approach, which does not account for the interplay between particle export and remineralisation in surface and subsurface layers.

Therefore, our model experiments show a lower sensitivity of global export production on biogeochemical parameters or
circulation than the previous studies. Some part of this difference could be explained by the large variation in physical model setup (in the study by Schmittner et al., 2005), or by the very different structure of the biogeochemical model applied (Najjar et al., 2007; Kwon et al., 2009). The large sensitivity of export production in the study by Najjar et al. (2007) also reflects on the range of deep particle flux (as diagnosed from export production times 0.052), which is almost ten times higher than in the present study.

Our experiments suggest that even though circulation does play a large role for export production, deep particle flux is mainly determined by parameter $b$. As a consequence, the sensitivity of export production to circulation noted by Najjar et al. (2007) and in the present study does not necessarily imply an equally large sensitivity of deep particle flux (and resulting remineralisation and deep oxygen consumption) on physical model features, which is somehow in contrast to the conclusions drawn by Najjar et al. (2007).





## 4.3   The effect of model complexity

To summarise, in all cases studied here biogeochemical parameter optimisation can help to narrow down the model uncertainty induced by circulation and biogeochemical parameters. Even if parameters optimised in one circulation are later transferred to a different circulation the resulting spread is mostly around 50% that of the model intercomparison presented by Bopp et al. (2013). However, that study included models that diverged not only in physics, but also in the biogeochemical structure, which might introduce another source of variability. To have a closer look at this we finally contrast the results of model MOPS, which we consider as a model of intermediate complexity, with an equivalent optimisation of a much simpler model "RetroMOPS" presented by Kriest (2017).

RetroMOPS is a four-component model that simulates only phosphate, nitrate, oxygen and dissolved organic phosphorus, but includes the same structural form for particle flux and remineralisation as model MOPS. When coupled to MIT28, and optimised against the same data set and misfit function presented above, the performance and global fluxes of RetroMOPS are very similar to MOPS (in the same circulation; Table 3 and Kriest, 2017)). For primary production the difference between RetroMOPS and MOPS is about as large as when MOPS is simulated with different parameter sets within a given circulation. One reason for this is the fact that the optimisation of RetroMOPS by Kriest (2017) aimed only at the parameters related to particle sinking and remineralisation, but not at parameters related to phytoplankton growth and loss terms; an additional optimisation of these parameters may likely have produced smaller differences.

Therefore, after optimisation a simple model can perform quite well with respect to large-scale biogeochemical quantities, in agreement with earlier findings (Kriest et al., 2012; Kwiatkowski et al., 2014; Galbraith et al., 2015), illustrating the benefit of parameter optimisation: on the one hand, optimisation allows for a "fair" comparison of models of different complexity (after each model has been tuned to match some desired quantity best); on the other hand it can also support model development, by helping to search for the best parameter set.

## 5   Conclusions

Optimisation of a global biogeochemical ocean model coupled to five different circulations achieved a good fit to observed nutrients and oxygen with partly different biogeochemical parameters. We identified three parameters that depend significantly on characteristic features of circulation, as summarised in Figure 13. Areas of deep ventilation in the North Atlantic and in the Southern Ocean determine how much oxygen is supplied to the ocean via air-sea gas exchange and subsequent mixing. As a consequence, optimisation of the model in circulations with vigorous ventilation triggers a high oxygen demand of remineralisation during optimisation. Fast turnover and mixing of NADW, as expressed through ideal age, affects the parameter responsible for the timescale and vertical extent of remineralisation of sinking particles. Here, models characterised by relatively young waters in the NADW adjust to deeper sinking and remineralisation, with consequences for the large scale distribution of phosphate. Finally, the combined outcrop area of SAMW and AAIW determines the optimal maximum rate of nitrogen fixation. This may be explained with the role of the Southern Ocean as "mixer" for waters of different origin and nitrogen deficit (Tuerena et al., 2015). The extent and properties of waters originating from this region (and their fixed nitrogen





deficit), in conjunction with denitrification within the OMZs then set the stage for the competitive advantage of cyanobacteria in tropical and subtropical waters. Models with a small outcrop area of these waters benefit from a low maximum rate of nitro-
gen fixation, and vice versa. In conclusion, when aiming for a good fit to large-scale biogeochemical quantities, key properties of the underlying circulation model should be considered; depending on region and tracer of interest, these key properties may, however, be different from those of our study.

Cross-validation experiments showed that the optimal parameters could be swapped between the different circulations to a limited extent. Parameters affect biogeochemical model performance in different ways: while the stoichiometric demand of
oxygen during remineralisation affects, for example, the ocean oxygen inventory, the particle flux parameter $b$ determines the large-scale distribution of nutrients, in line with earlier studies (Bacastow and Maier-Reimer, 1991; Kwon and Primeau, 2006; Kriest et al., 2012).

Compared to other intercomparisons of global coupled models tuned more subjectively, our overall variation of biogeochemical key properties is at least 50% smaller, with different contributions from circulation and biogeochemical parameters.
For example, export production seems to be mainly determined by circulation, while deep particle flux is determined almost entirely by the particle flux parameter $b$. Other biogeochemical diagnostics are affected more or less equally by circulation and biogeochemical parameters. Finally, OMZ volume is very sensitive to changes in circulation and biogeochemical parameters, and varies most strongly across all model experiments.

However, models considered in global intercomparisons usually differ also in their biogeochemical structure and complexity.
Our experiments suggest that after optimisation the differences due to model structure are much smaller than those due to model parameters or circulation. This indicates that a simpler model can perform as well as a more complex model (with respect to the metrics and diagnostics applied here), similar to the results obtained by Kriest et al. (2012), Kwiatkowski et al. (2014) and Galbraith et al. (2015). It also illustrates how biogeochemical parameter optimisation can aid model development: whenever new components or parameterisations are introduced to a global model, this new model has to be tuned in order to match
observations on global or regional scales. Often, the choice of appropriate parameters is not easy, and requires extensive testing and sensitivity analysis. Automatic (algorithmic) optimisation can make calibration more efficient, simplifying the search for a good model match to observed quantities. For a given misfit function it can also support decisions about the necessary level of model complexity.

Therefore, our experiments suggest that global biogeochemical ocean models benefit from optimisation, even if this was
carried out in a circulation differing from that of the "target" circulation. However, to date there is no guarantee that a model showing a good fit to observed quantities in steady state (i.e., when simulated in a preindustrial or present day, climatological physical forcing) will exhibit the correct response when applied to transient scenarios reflecting future climate change. As shown here and in earlier studies (Cocco et al., 2013; Cabre et al., 2015; Löptien and Dietze, 2019), OMZs seem to be particularly sensitive to both biogeochemical and physical parameters. Accounting for the match between simulated and observed
OMZs during optimisation can reduce the model spread in steady state (Sauerland et al., 2019). Extending the simulation of optimal models to future states could then inform us about their sensitivity to changes in circulation and forcing, and may pro-





vide a better constraint on their uncertainties. Thus, the study presented here serves as a first step to unravel the uncertainties associated with the divergence of global biogeochemical model performance and uncertainty.

*Code and data availability.* Model code employed for the experiments presented in this paper, as well as model output are available at
https://data.geomar.de/thredds/catalog/open_access/kriest_et_al_2020_bg/catalog.html (last access: 6 Feb 2020). The TMM source code is available from the TMM repository at https://github.com/samarkhatiwala/tmm (last access: 6 Feb 2020).

*Author contributions.* IK designed the study, carried out and analysed the experiments. KK provided TMs, forcing and ideal age for UVic experiments. WK provided ideal age for MIT28 and ECCO experiments. All authors discussed the results and wrote the manuscript.

*Competing interests.* The authors declare that they have no conflict of interest.

*Acknowledgements.* This work is a contribution to the DFG-supported project SFB754 and to BMBF joint project PalMod (FKZ 01LP1512A). Parallel supercomputing resources have been provided by the North-German Supercomputing Alliance (HLRN). We thank the Biogeochemical Modelling Group at GEOMAR for discussions and many helpful suggestions. The authors wish to acknowledge use of the Ferret program of NOAA's Pacific Marine Environmental Laboratory for analysis and graphics in this paper.



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





**Table 1.** Results of optimisations, expressed as the minimum misfit $J^*$ (the target of optimisation), bias of global average nitrate and oxygen, bias of OMZ volume (for OMZs defined by $O_2 < 50$ and $O_2 < 80$ mmol m$^{-3}$), optimal parameters and their uncertainties. To determine parameter uncertainty, we defined the group $\Omega$ of model simulations whose misfit $J_k$ deviates less than 1% from the optimal misfit $J^*$ ($J_k/J^* - 1 \leq 0.01$). For each parameter the first value gives the optimal parameter, averaged over the last generation, and the range of all individuals of $\Omega$ is given in squared brackets.

| | MIT28* | UHigh* | U20* | U17.5* | ECCO* |
|---|---|---|---|---|---|
| Ocean model: | MIT28 | UVic | UVic | UVic | ECCO |
| Horizontal resolution: | $2.8° \times 2.8°$ | $3.6° \times 1.8°$ | $3.6° \times 1.8°$ | $3.6° \times 1.8°$ | $1.0° \times 1.0°$ |
| Vertical resolution: | 15 | 19 | 19 | 19 | 23 |
| *Global metrics:* | | | | | |
| $J^*$ | 0.439 | 0.401 | 0.431 | 0.489 | 0.366 |
| Bias NO$_3$ (mmol m$^{-3}$) | -0.27 | 0.68 | 0.30 | 0.52 | 0.28 |
| Bias O$_2$ (mmol m$^{-3}$) | 3.03 | 1.63 | 3.56 | 5.29 | 0.08 |
| Bias OMZ Vol. 50 ($10^{15}$ m$^3$) | -19.0 | -5.7 | 3.1 | 12.6 | -23.2 |
| Bias OMZ Vol. 80 ($10^{15}$ m$^3$) | -44.0 | 16.8 | 11.7 | 22.7 | -40.7 |
| *Parameters [Range]* | | | | | |
| $b$ | 1.39 [1.3-1.5] | 1.27 [1.2-1.3] | 1.40 [1.3-1.5] | 1.44 [1.3-1.6] | 1.46 [1.4-1.5] |
| $R_{-O2:P}$ | 173.7 [166-178] | 161.5 [158-166] | 161.5 [158-167] | 167.0 [159-175] | 151.1 [150-154] |
| $\mu_{\text{NFix}}$ | 1.19 [1.1-1.8] | 2.98 [2.1-3.0] | 1.00 [1.0-1.4] | 1.98 [1.5-3.0] | 2.29 [1.5-3.0] |
| $DIN_{\text{min}}$ | 15.8 [12.2-16.0] | 16.0 [6.2-16.0] | 15.9 [7.0-16.0] | 16.0 [12.0-16.0] | 16.0 [10.2-16.0] |
| $K_{\text{DIN}}$ | 32.0 [17.2-32.0] | 26.5 [17.4-31.4] | 31.9 [25.8-32.0] | 32.0 [21.3-32.0] | 23.1 [21.7-32.0] |
| $K_{\text{O2}}$ | 1.01 [1.0-8.5] | 13.58 [2.1-14.3] | 7.15 [1.0-8.0] | 5.62 [1.5-16.0] | 1.07 [1.0-16.0] |





**Table 2.** Correlation coefficient $r$ for the regression of three optimal model parameters against physical diagnostics. The outcrop area of dense waters in the northern ($> 40°$N) and southern ($> 40°$S) hemisphere is given for two different density intervals. Outcrop area for deep mixing in the North Atlantic ($> 40°$N) and Southern Ocean ($> 40°$S) is given for two different criteria of maximum deep mixing (200m and 400m). Mean water mass age is given for four different water masses. See text for further details. Significant correlations ($p < 0.05$) are denoted by bold face.

| Parameter | Physical diagnostic | | | |
|---|---|---|---|---|
| | Area MLD North Atlantic | | Area MLD Southern Ocean | |
| | 200m | 400m | 200m | 400m |
| $R_{-\mathrm{O2:P}}$ | **0.916** | **0.950** | 0.875 | 0.465 |
| $\mu_{\mathrm{NFix}}$ | -0.492 | -0.405 | -0.332 | -0.263 |
| $b$ | -0.280 | -0.425 | -0.499 | -0.210 |
| | Area Outcrop North | | Area Outcrop South | |
| | 26.5-27.5 | >27.5 | 26.5-27.5 | >27.5 |
| $R_{-\mathrm{O2:P}}$ | -0.329 | 0.460 | -0.641 | 0.792 |
| $\mu_{\mathrm{NFix}}$ | 0.449 | -0.361 | **0.890** | -0.746 |
| $b$ | -0.287 | 0.050 | -0.538 | 0.202 |
| | Ideal Ages | | | |
| | NADW | CDW | NPDW | ETP |
| $R_{-\mathrm{O2:P}}$ | -0.358 | 0.273 | 0.787 | -0.266 |
| $\mu_{\mathrm{NFix}}$ | -0.083 | -0.380 | -0.857 | -0.070 |
| $b$ | **0.914** | 0.136 | 0.069 | 0.334 |





**Table 3.** Mean (across all experiments and optimal models) and range of variation of global biogeochemical model properties and fluxes across different parameter sets (circulation constant; $\Delta$ Par), and different circulations (parameters constant; $\Delta$ Circ), as well as across the five different optimal models MIT28*, ECCO*, uHigh*, U20* and U17.5* ($\Delta$ Opt). $\Delta$ Mod shows the difference between MIT28* of this study and model RetroMOPS of Kriest (2017). Observed oxygen and OMZ volume from Garcia et al. (2006b, mapped onto ECCO grid); observed global flux ranges derived from estimates by Carr et al. (2006, primary production), Dunne et al. (2007, export production), Lutz et al. (2007, export production; radiogen. calib.), Honjo et al. (2008, mean particle flux), Guidi et al. (2015, particle flux), Eugster and Gruber (2012, median fixed nitrogen loss) and Somes et al. (2013, fixed nitrogen loss of best data-constrained model).

| | Mean (All) | Mean (Opt) | $\Delta$ Mod | $\Delta$ Par | $\Delta$ Circ | $\Delta$ Opt | $\Delta$ All |
|---|---|---|---|---|---|---|---|
| Global mean $O_2$ (observed: 174.17 mmol m$^{-3}$) | | | | | | | |
| This study | 177.2 | 177.6 | 0.1 | 24.1 | 24.3 | 6.8 | 43.0 |
| Bopp et al. (2013) | | | | | | | 95.0 |
| OMZ Volume (50 mmol m$^{-3}$; observed: $57.0 \times 10^{15}$ m$^3$) | | | | | | | |
| This study | 54.9 | 52.1 | 0.7 | 39.4 | 55.4 | 39.0 | 67.2 |
| Bopp et al. (2013) | | | | | | | 212.5 |
| OMZ Volume (80 mmol m$^{-3}$; observed: $119.1 \times 10^{15}$ m$^3$) | | | | | | | |
| This study | 122.3 | 112.8 | 0.6 | 112.0 | 119.8 | 73.3 | 145.9 |
| Bopp et al. (2013) | | | | | | | 328.9 |
| Fixed N loss (observed estimates: 52-76 Tg N y$^{-1}$) | | | | | | | |
| This study | 67.7 | 67.4 | 1.9 | 35.1 | 33.8 | 17.4 | 44.9 |
| Somes et al. (2013)[§] | | 53.6 | | | | | |
| Primary Production (observed estimates: 40-60 Pg C y$^{-1}$) | | | | | | | |
| This study | 47.4 | 47.2 | 9.1 | 8.17 | 12.27 | 5.65 | 18.95 |
| Schmittner et al. (2005)[§] | | | | 20.1 | 24.5 | | 48.5 |
| Seferian et al. (2013) | | | | | 8.64 | | |
| Bopp et al. (2013) | | | | | | | 47.8 |
| Export Production (observed estimates: 4.6-9.6 Pg C y$^{-1}$) | | | | | | | |
| This study | 6.86 | 6.86 | 0.23 | 0.49 | 1.08 | 1.05 | 1.20 |
| Schmittner et al. (2005)[§] | | | | 2.2 | 8.4 | | 11.2 |
| Najjar et al. (2007) | | | | | 10 | | |
| Kwon et al. (2009)[†] | | | | $\approx$5 | | | |
| Seferian et al. (2013) | | | | | 3.0 | | |
| Bopp et al. (2013) | | | | | | | 3.2 |
| Sedimentation (2000m) (observed estimates: 0.33-0.43 Pg C y$^{-1}$) | | | | | | | |
| This study | 0.36 | 0.36 | 0.02 | 0.16 | 0.06 | 0.16 | 0.19 |
| Najjar et al. (2007)[‡] | | | | | 0.52 | | |

[§] experiments 1-3; [§] For $\Delta$Circ we have omitted the experiment with $K_b = 0$ of the study by Schmittner et al. (2005), and refer only to experiments 2, 8 and 12. For $\Delta$Par we report the difference between experiments 12 and 13, and for $\Delta$All the maximum spread across experiments 2 to 13. [†] We refer to Figure 2b of Kwon et al. (2009), but consider only a range of $1.1 \leq b \leq 1.4$ for $\Delta$Par, to be comparable to our range of $b$. [‡] Calculated from export production $\times (75/2000)^{0.9} = 0.052$.

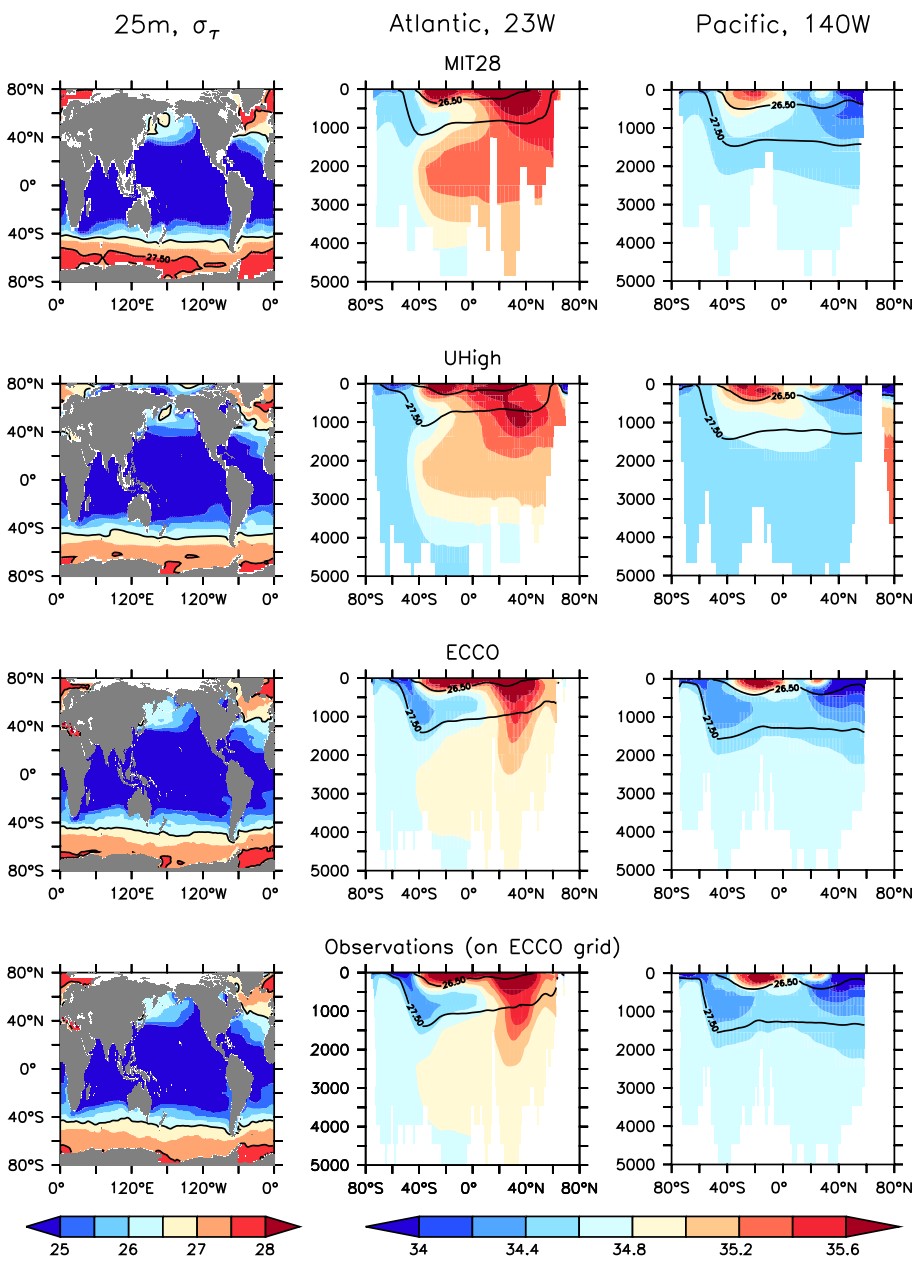

**Figure 1.** Density ($\sigma_\Theta$) at 25 m (left panels), and salinity along sections at 23°W and 140°W (middle and right panels), of forcing from (top to bottom) MIT28, UHigh and ECCO circulation. The lower panel shows observations mapped onto ECCO geometry. Density has been derived from annual mean potential temperature and salinity. Contour lines highlight isopycnal of $\sigma_\theta = 26.5$ and $\sigma_\theta = 27.5$.



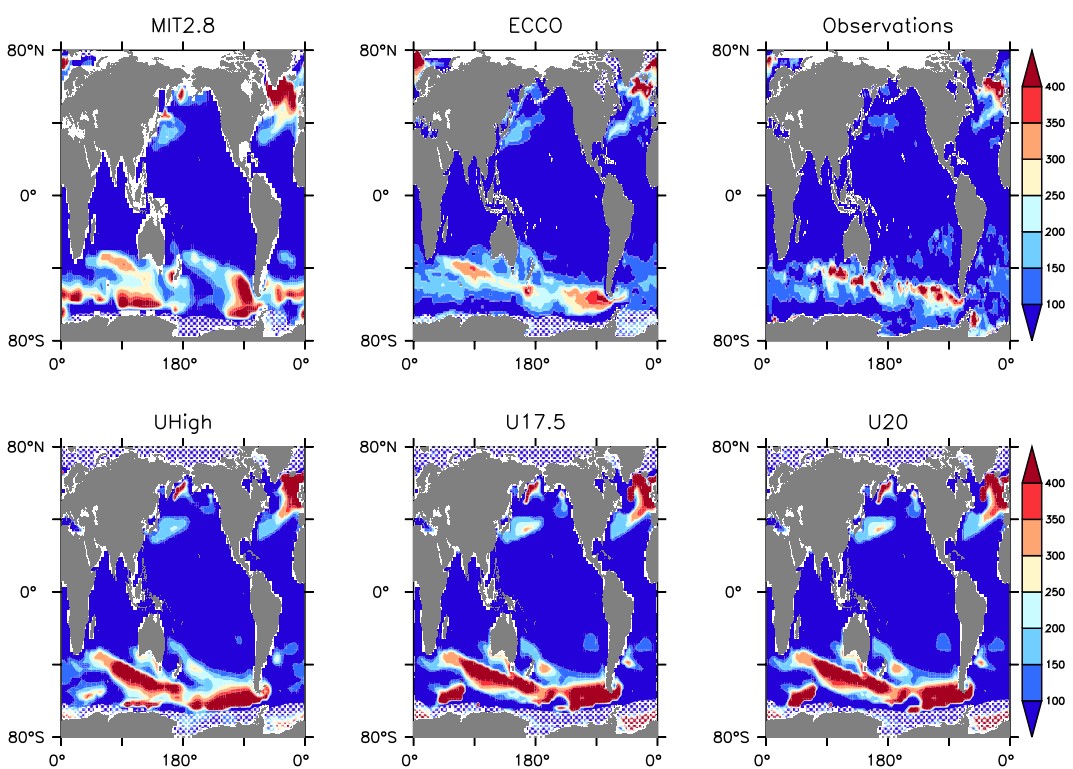

**Figure 2.** Annual maximum mixed-layer depths of models and observations (gridded onto ECCO model geometry). Mixed-layer depths have been determined from a density difference of $\sigma_\theta = 0.03$ (density calculated from monthly mean temperature and salinity). Square pattern in high latitudes denotes regions with an ice coverage of at least 50%.




**Figure 3.** Simulated ideal age (years) averaged over 1500-2500 m in the North Atlantic (NADW; left panels), and as zonal mean for the Atlantic (mid panels) and the Pacific (right panels). Boxes indicate regions of NADW, NPDW, CDW and ETP (see Fig. S2 for detailed plot).





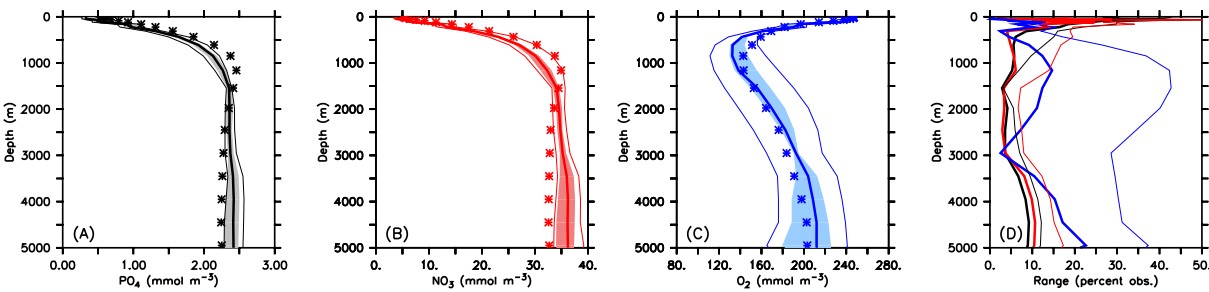

**Figure 4.** Global mean vertical profiles of phosphate (A), nitrate (B) and oxygen (C). Thin lines denote range across all 25 model experiments, shaded areas the range across the five optimal model experiments and the thick lines the average of optimal model experiments. Stars denote observed profiles. The right panel (D) shows the range (as percent of observations) across optimal models as thick lines and the range across all model experiments as thin lines. Black lines: phosphate; red lines: nitrate; blue lines: oxygen. Prior to averaging, all model have been regridded onto the ECCO grid, using nearest neighbour filling in the vertical, and linear interpolation horizontally.

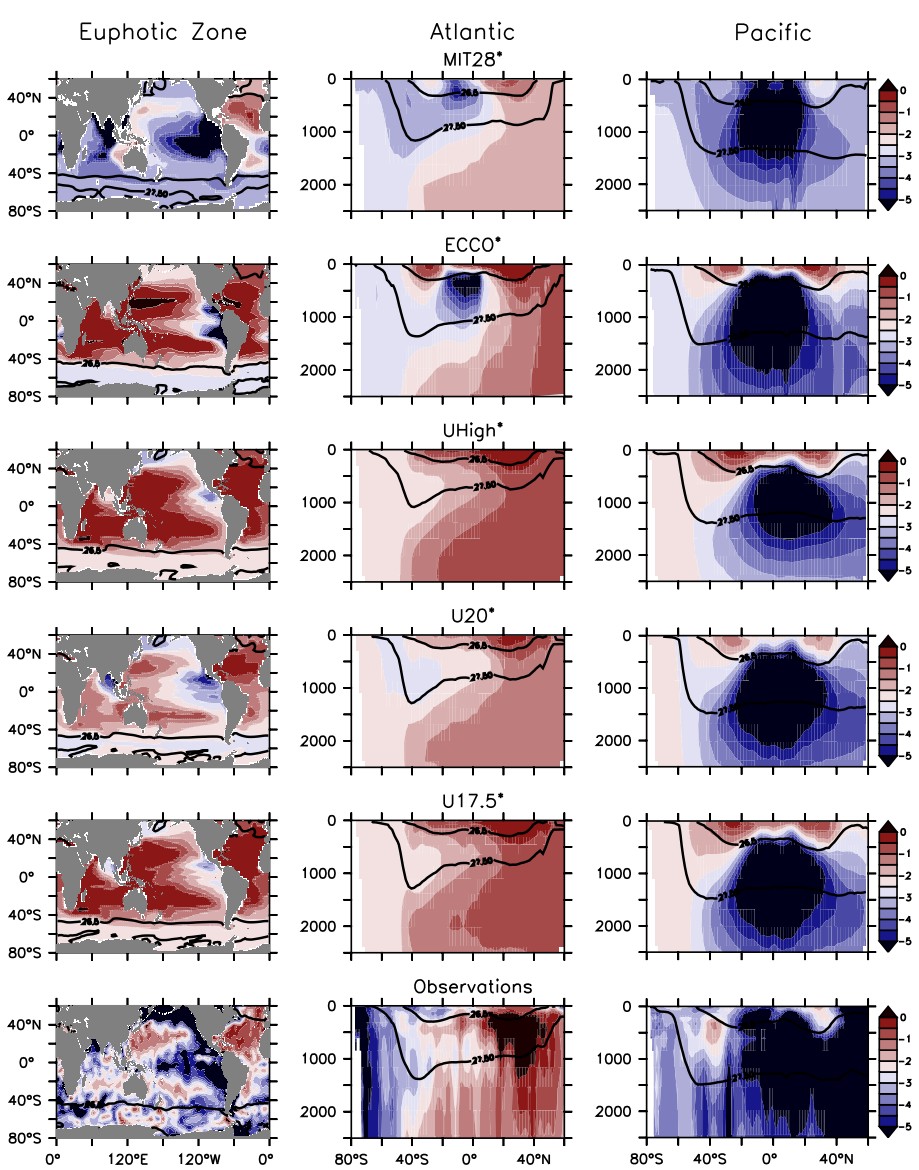

**Figure 5.** Excess nitrate N* (N* = NO$_3$ − 16 × PO$_4$) of optimal model configurations, averaged over the euphotic zone (left panels), and of zonal mean nitrate and phosphate in the Atlantic and Pacific (middle panels and right panels, respectively). Lines denote density of $\sigma_\theta = 27$ and $\sigma_\theta = 27.5$.



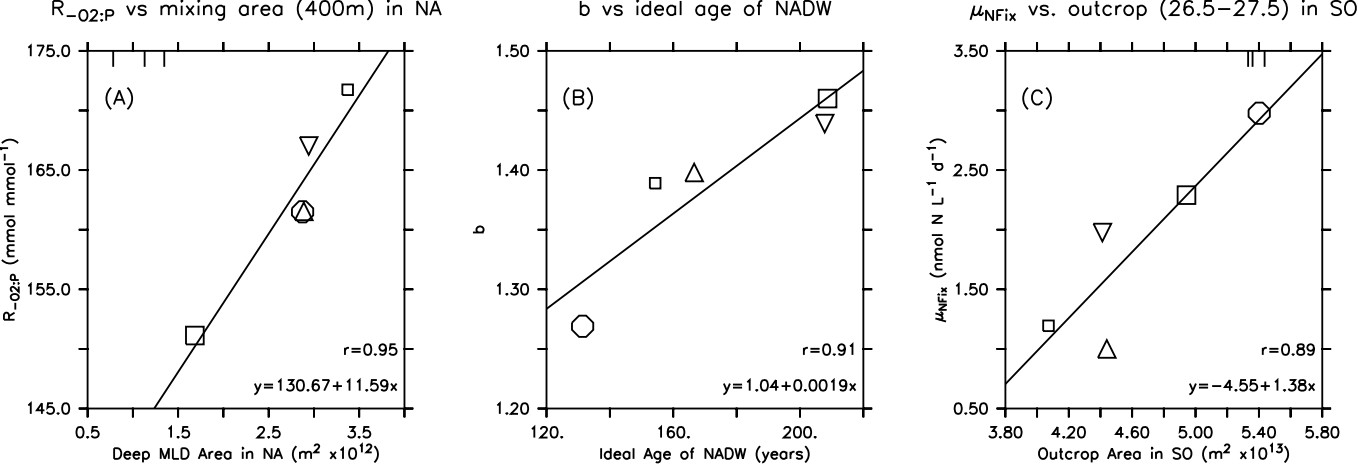

**Figure 6.** Optimal parameters for which the correlation of Table 2 is significant at $p < 0.05$, plotted against physical diagnostics. Symbols indicate the different model optimisations. Small squares: MIT28*. Large squares: ECCO*. Circles: UHigh*. Triangles: U20*. Inverted triangles: U17.5*. Lines denote the regression of optimal parameters against the respective circulation diagnostic. Vertical bars at the upper plot boundary indicate observed diagnostic.

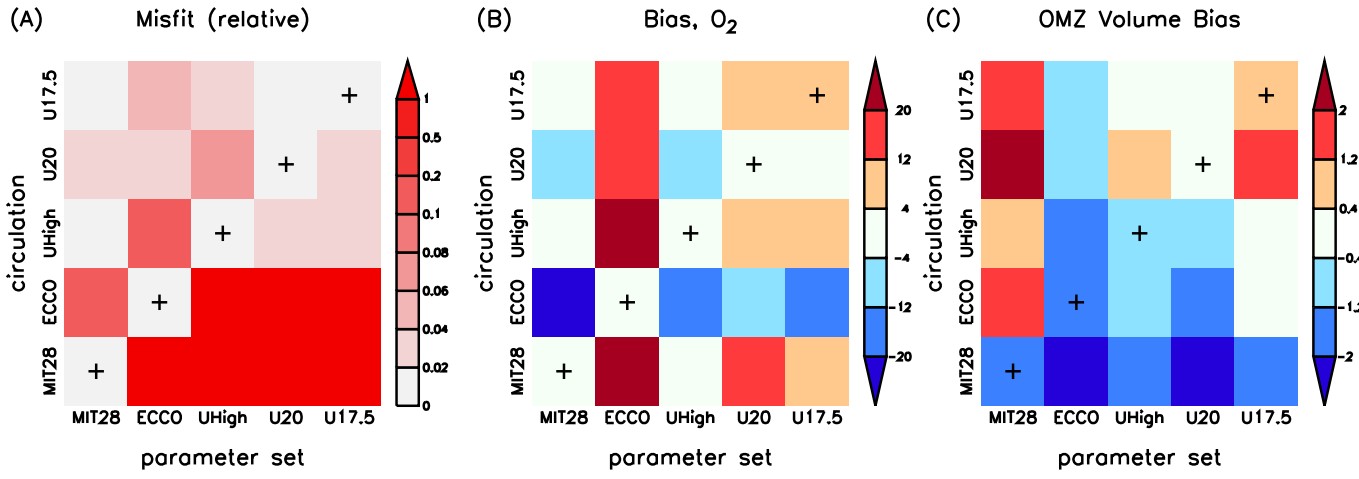

**Figure 7.** Three different model diagnostics for all cross-validation experiments with parameter set $i$ and circulation $j$: (A) normalised misfit $J_{i,j}/J_j^* - 1$, where $J_j^*$ is the lowest misfit for each circulation $j$. (B) oxygen bias (model minus observation, [mmol m$^{-3}$]). (C) OMZ volume (as percent of total ocean volume) bias. OMZs are defined by 50 mmol m$^{-3}$. The x-axis denotes the optimal parameter sets of MIT28*, ECCO*, UHigh*, U20*, U17.5* and the y-axis the circulation. Pluses along the main diagonal indicate the optimal model simulation for each circulation ($i = j$), i.e., MIT28*, ECCO*, UHigh*, U20*, U17.5*.





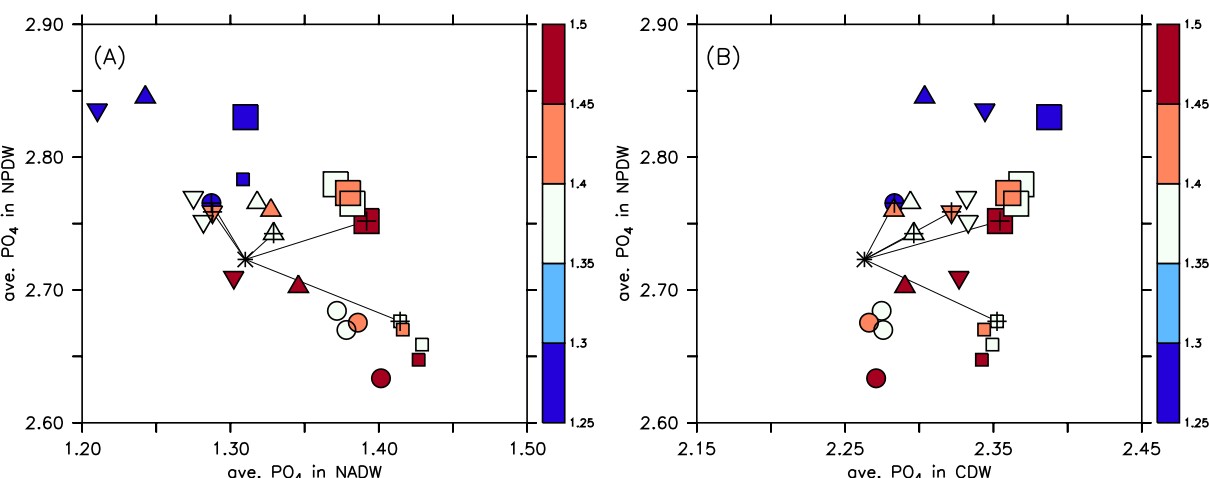

**Figure 8.** Average phosphate in the northern North Pacific Deep Water (NPDW; between 0°N and 60°N, and 1500-5000 m; see Text) plotted against average phosphate in the northern North Atlantic Deep Water (NADW; between 0°N and 60°N, and 1500-2500m; panel (A)) or Circumpolar Deep Water (CDW; south of 45°S, 1500-5000 m; panel (B)), of all 25 model experiments. Small squares: MIT28 circulation. Large squares: ECCO circulation. Circles: UHigh circulation. Triangles: U20 circulation. Inverted triangles: U17.5 circulation. The colour indicates the value of parameter $b$. Pluses indicate the optimal parameter set. Stars indicate observed values. Thin black lines extending from the stars denote the distance between observation and each optimal model configuration.

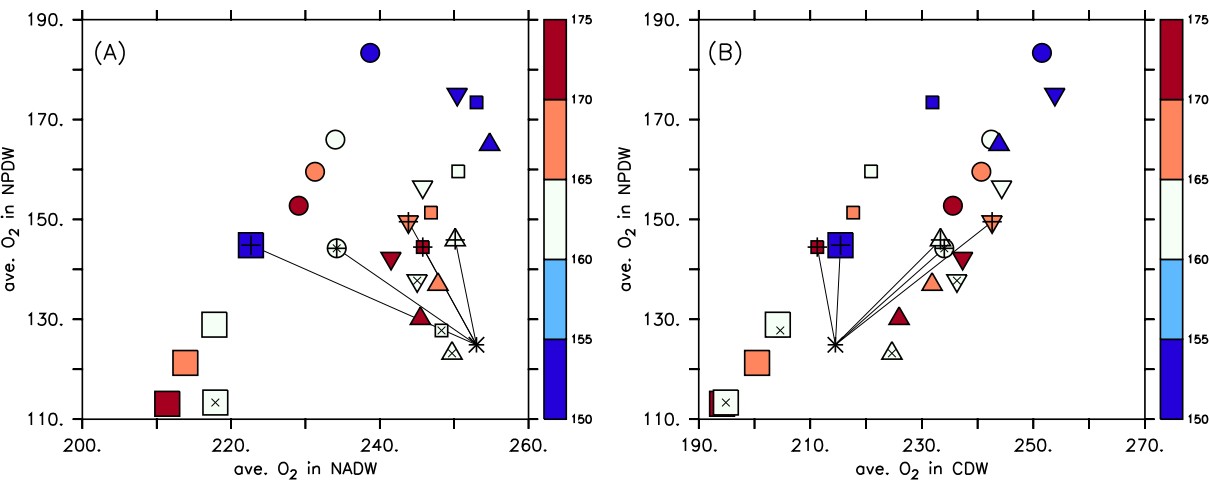

**Figure 9.** As Figure 8, but for average oxygen and parameter $R_{-O2:P}$ (colour scale).

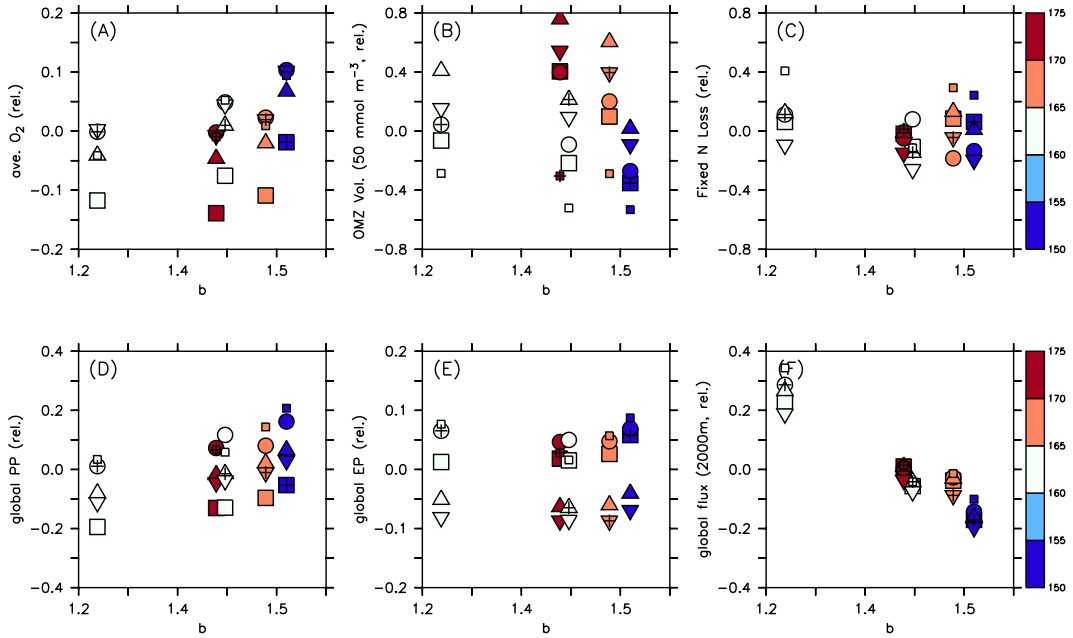

**Figure 10.** Normalised biogeochemical diagnostics plotted against parameter $b$: (A) Global average oxygen, (B) OMZ volume defined by concentrations $< 50$ mmol m$^{-3}$, (C) global fixed N loss, (D) global primary production, (E) export production and (F) organic particle flux at 2000 m. All diagnostics $X$ expressed as relative deviation to the mean of the five optimal model simulations ($X_{i,j}/\bar{X}_i^* - 1$), where $j$ and $i$ denote different combinations of circulation $j$ and parameter set $i$, and $\bar{X}_j^*$ the average of all optimal model configurations (see Table 3 for values). Colour denotes value of parameter $R_{-O2:P}$. Symbols denote circulation: Small squares: parameter set of MIT28$^*$. Large squares: ECCO$^*$. Circles: UHigh$^*$. Triangles: U20$^*$. Inverted triangles: U17.5$^*$. Optimal model configurations are indicated by pluses.

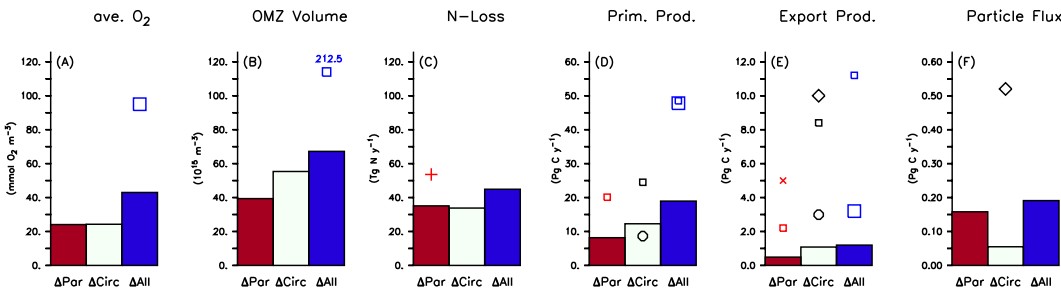

**Figure 11.** Effect of variation in biogeochemical parameters ($\Delta$Par), circulation ($\Delta$Circ) and across all model experiments ($\Delta$Circ) on (A) global average oxygen, (B) OMZ volume defined by a concentration of $< 50$ mmol m$^{-3}$, (C) global fixed nitrogen loss, (D) global primary production, (E) export production and (F) organic particle flux at 2000 m, as listed in Table 3. Symbols denote values from Bopp et al. (2013, large squares), Schmittner et al. (2005, small squares), Seferian et al. (2013, circles), Najjar et al. (2007, diamonds), Somes et al. (2013, plus) and Kwon et al. (2009, cross).





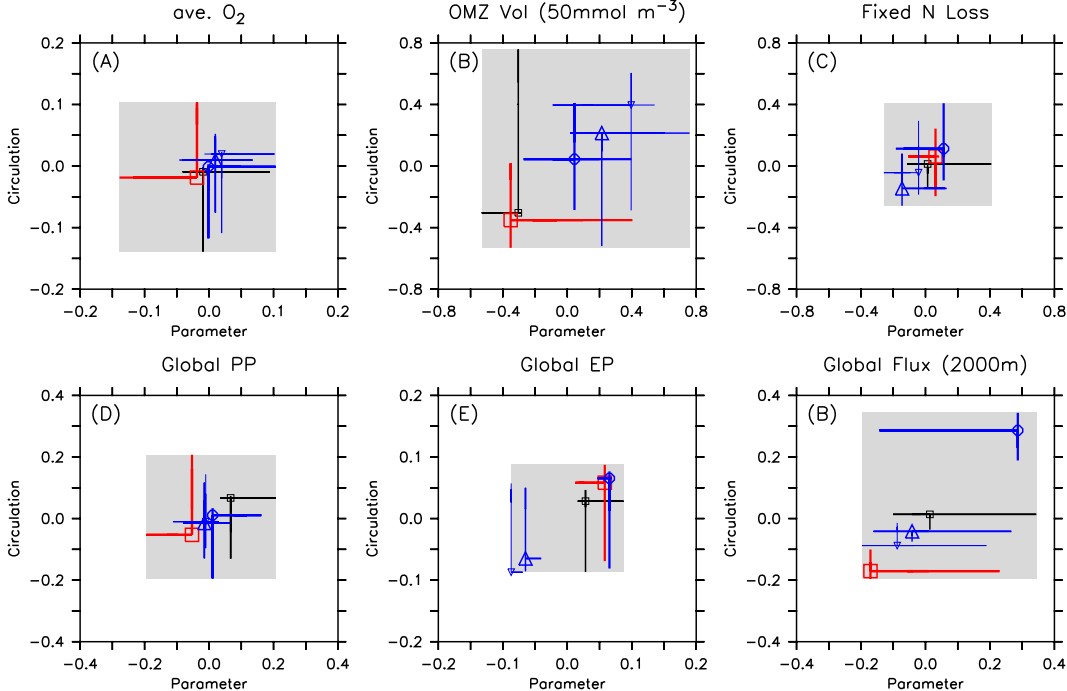

**Figure 12.** Effect of variation in circulation and biogeochemical parameters on normalised diagnostics. (A) global average oxygen, (B) OMZ volume defined by a concentration of $< 50$ mmol m$^{-3}$, (C) global fixed nitrogen loss, (D) global primary production, (E) export production and (F) organic particle flux at 2000 m. All diagnostics $X$ are expressed as relative deviation to mean of the five optimal model simulations $(X_{i,j} / \bar{X}_j^* - 1)$, where $j$ and $i$ denote different combinations of circulation $j$ and parameter set $i$, and $\bar{X}_j^*$ is the average of all optimal model configurations (see Table 3 for values). Each panel shows the range of the normalised diagnostic when the parameter set is kept constant and circulation varied (vertical lines), and when circulation is kept constant and the parameter set is varied (horizontal lines). Symbols denote the optimal model configuration. Colour denotes circulation and optimal parameter set. Black: circulation MIT28 or parameter set of MIT28$^*$. Thick blue: UHigh/UHigh$^*$. Medium blue: U20/U20$^*$. Thin blue: U17.5/U17.5$^*$. Red: ECCO/ECCO$^*$. Symbols denote optimal model configurations. The grey-shaded area shows total variation across all 25 model experiments, corresponding to $\Delta$ All in Table 3. Note that the maximum variation due to parameter within each circulation ($\Delta$Par of Table 3) is given by the longest horizontal line in each panel. Likewise, the maximum variation due to circulation within each parameter set ($\Delta$Circ of Table 3) is given by the longest vertical line.





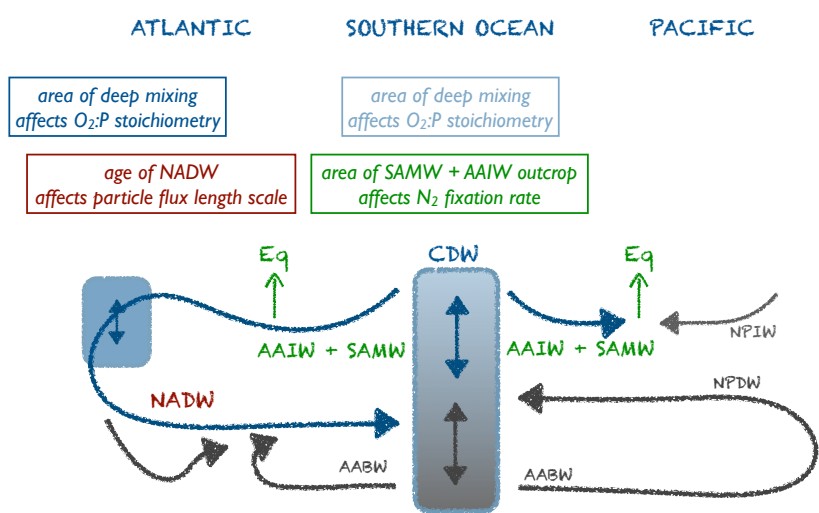

**Figure 13.** Cartoon depicting the simplified large scale circulation pattern of the Atlantic and Pacific Ocean, and the dependence of three biogeochemical parameters $R_{-O2:P}$, $b$ and $\mu_{\mathrm{NFix}}$ of physical properties.