# Peer review of "One size fits all? - Calibrating an ocean biogeochemistry model for different circulations"

_Biogeosciences, 2020_

## Referee Comment (RC1) · Anonymous Referee #1 · 10 Mar 2020

Kriest et al., calibrate a single ocean biogeochemical model with five different ocean circulation models (in the form of transport matrices) and explore how the model-observation misfit varies as a function of the suite of calibrated biogeochemical parameters and of various metrics of ocean circulation. The authors find consistent relationships between metrics, parameter values and model-observation misfits despite differences in calibrated parameter values. The authors then explore how each calibrated parameter set performs with an alternative circulation model finding that calibration in other models, such as coarse-resolution models, can still reduce the misfit.

Overall, this is a very well-designed set of model experiments and a very thorough analysis. The authors have done an impressive job of analysing and communicating a very complex set of results! The study raises a lot of interesting and important scientific

questions about the calibration and complexity of biogeochemical models that are significant to the wider biogeochemical modelling community. Some of the results could be specific to the biogeochemical model used but the authors openly discuss this. The study itself is sufficiently self-contained that I think this is not a problem (and is a necessity given the complexity of the results). I recommend very minor revisions, mainly to improve the quality of the figures which otherwise hinder the visual interpertation.

General Comments:

Some of the figures were a little hard to interpret due to the large number of model set-ups. Figures 6, 8, 9, 10 and 12 would really benefit from a legend or key to more quickly pick out which model set-up is which (a legend on the first relevant plot that is referenced for subsequent plots for example). Interpreting the plots with reference to the text was difficult because of this.

Specific Comments:

Section 2.7: I found this section a little unclear due to the discussion about the previous optimisations. It would help to focus on the parameters varied in this study based on the second 2017 optimisation and mention more briefly that the plankton parameters were the same as the first 2017 optimisation.

Line 347: "MOPS coupled to UVic circulation is more robust with respect to changes in parameters." : the sentence meaning is unclear, does this mean that the calibrated parameters are similar across the three UVic circulation used? (In comparison to ECCO or MIT28).

Line 349: "...the large impact of oxygen on the misfit function...": could you elaborate briefly why this is the case here.

Lines 399 – 413: this analysis assumes that the interactions between circulation, biogeochemistry parameters and the misfit are linear and additive? Figure 11 suggests that this might not be the case as the delta_par and delta_circ bars do not sum to the

delta_all bar. The analysis in this form is fine (and considering non-linear interactions would not be easy!) but I think this assumption should be mentioned.

Lines 453 – 454: "...it prevents fast settling of organic particles out the euphotic zone" is a little unclear. Does this mean there is effectively an increased residence time of particles in the euphotic zone which equates to a larger fraction of particles being remineralised before reaching the ocean interior? Is there also an impact of the plankton model in this instance, e.g., changes in zooplankton grazing?

Lines 459 – 461: "long term storage of nutrients and carbon will, to a large extent, depend on the prescribed particle flux profile" – the air-sea balance of $CO_2$ might depend on circulation more than nutrients to the gas exchange component, similarly to the arguments made about $O_2$ previously.

Line 469 - 474: There should be a caveat that these findings are for MOPS specifically.

Figure 4D is very hard to interpret due to the colour contrasts and placement/combination of lines. The panel is not explicitly mentioned in the text so I would suggest to move the figure to supplementary or separate into more panels to make it clearer.

---

## Referee Comment (RC2) · Anonymous Referee #2 · 19 Mar 2020

In this study, Kriest et al. investigate the role of different circulation realisations on their impact on the optimisation of the parameters of a biogeochemical model. They identify which parameters are most sensitive, and attribute this sensitivity to key physical aspects of the circulation states, principally relating to the scale of mixing and ventilation. They also "port" optimised parameter sets between circulation states to attempt to quantify how transferrable optimisations of models are. In particular, they note that optimisations at low spatial resolution circulations can help in simulations at higher resolution.

This manuscript deals with a significant problem affecting marine biogeochemical modelling. Essentially, because of the computational cost of full 3D simulation, especially at high resolution, we typically make do (whether this is acknowledged or not) with

suboptimal parameterisations (and I say this as a suboptimal modeller!). Even where optimisation is attempted, its specificity to an ocean circulation state, and its portability between such states, is unexplored and unknown. As such, this study fits a definite need in the community.

Overall, this is a really well-designed and executed study. The authors have done a great job covering the main angles, and done so with a good spread of model circulations. In particular, I like the relating of model parameter values to circulation properties – that's valuable for modellers looking to compare this work with their own.

I have no significant criticisms of the work, and have only minor corrections or suggestions to make. My recommendation is publication after minor corrections.

Specific comments:

Line 28: the use of two maxima here is a little confusing; can this be clarified at all?; a range of 180% might even be explicable by such mismatch of OMZ definitions, but I doubt that's what's happened here

Line 38: "by hand" is, in part, related to keeping the number of tuning simulations down; were a model to be inexpensive to run, automatic tuning sampling a large number of parameter sets would likely be preferred

Line 59: "seems to be" -> "has proven to be"

Lines 60-69: a very nice framing of the problem; thanks

Line 81: it seems remiss not to include even a sentence or two explaining what the TMM is (or what it comprises); it would spare your readers to present something here

Line 87: move this domain information into a table instead?

Line 104: interesting numbers are reported for this domain, but not the others; it would be good to have all, or at least some other common information across the TMs

Lines 109-110: what does "accurately represent" here mean?; is it in reference to the circulation strengths you mention, or something else?

Line 123: could a calculation of ventilation timescale (mean, max, and between basins) help separate out the differences between the TMs?

Line 129: "up to more than 800 years"; any idea why?; that does sound surprising at face value

Line 130: ah-ha - ages mentioned here, but perhaps these could be added to the table I mentioned before for clarity

Line 132: "tracer observations"?; do you mean radiocarbon?

Lines 137-138: this section feels like it could do with a sentence explaining how this information will be used later; however, it's certainly very helpful to elucidate how models might be good / bad

Line 169: per my earlier remark, how's Drake Passage transport in the models?; it has a relationship with the SO properties mentioned

Line 204: how is near steady state defined? (presumably in Kriest et al., 2017 ...)

Line 210: "many local optima"; good - this is a perennial problem with BGC models

Line 214: as these properties are tightly constrained in the real ocean (e.g. via the N:P ratio), is there an advantage to using all of them?; i.e. could N, O2 or P, O2 be sufficient?; something like carbon - which has a more plastic relationship with nutrients - might arguably be good too

Line 239: you could add the total number of simulations here to indicate the total computational load

Line 245: earlier, in relation to burial and riverine input, you suggest that the inventory is not actually fixed, and can drift by a few percent; which is right?

Line 284: why especially oxygen?

Line 290: so that readers (like this one) don't have to scramble back a few pages, perhaps reiterate the default values once when you mention changes here

Line 362: it's not *completely* independent when the model is optimised to oxygen concentration; although, I appreciate it's not a target

Line 428: should be "m3" not "m-3"; also, might want to contextualise with a percent of mean

Lines 639-640: "on the other hand ..." is a confusing point; what do you mean?; it seems to suggest that a "benefit of parameter optimisation" is "helping to search for the best parameter set"; that sounds not particularly profound; something instead about "necessary level of model complexity"?

Line 647: "through low ideal age"?

Table 1: mean ocean ventilation age from these different circulations might be an interesting metric; or some other relevant integral metric of circulation

Table 3: export production - I like this illustrating of gaps in previous work (which has been more opportunistic than this study)

Figures: a general point I'd make is that red/blue colour bars are usually for situations where a property (e.g. a delta) has a definite central point worth marking (e.g. zero); here they're used broadly, potentially skewing the reader's perspective

Figure 1: you may have tried already, but might delta plots be better? (i.e. show the observations as field values but models as differences from this)

Figure 2: as the observations are missing ice, I'd be inclined to skip it in the models as well

Figure 3: see general and Figure 1 comments

Figure 4: panel D is rather complicated and ugly

Figure 5: change to a much uglier palette; same red / blue issue

Figure 6: I really like this relating of parameter value to circulation property

Figures 8 and 9: I found these rather difficult to interpret, although I have no suggestions on how to change them

---

## Author Comment (AC1) · 6 Apr 2020

We thank referee 1 very much for his/her encouraging and helpful comments! Please find below our reply and answer to the referee's suggestions and comments (in blue).

General Comments: Some of the figures were a little hard to interpret due to the large number of model set-ups. Figures 6, 8, 9, 10 and 12 would really benefit from a legend or key to more quickly pick out which model set-up is which (a legend on the first relevant plot that is referenced for subsequent plots for example). Interpreting the plots with reference to the text was difficult because of this.

We have now added a symbol legend to Figure 6 and will refer to this in plots 8, 9, 10 and 12. Note that we will also increase the number of digits in the regression equations

(formerly truncated to 2-4).

Specific Comments: Section 2.7: I found this section a little unclear due to the discussion about the previous optimisations. It would help to focus on the parameters varied in this study based on the second 2017 optimisation and mention more briefly that the plankton parameters were the same as the first 2017 optimisation.

We found it somehow difficult to rephrase this by just mentioning the first optimisation by Kriest et al. (2017) in passing (and secondly), but suggest to rewrite this subsection to hopefully make it easier to understand the sequence of optimisations: *"The optimisations presented here are based upon two successive optimisations presented by Kriest et al. (2017) and Kriest (2017). Both studies applied model MOPS coupled to TMs derived from MIT28. The cost function, as presented in Equation 1 was calculated after a spin up of 3000 years. In the first optimisation, Kriest et al. (2017) optimised four parameters related to plankton growth and loss terms, together with $b$ and $R_{-O2:P}$. The optimal parameters of this first calibration led to a better agreement of simulated global biogeochemical fluxes to observations of primary and export production, zooplankton grazing, particle flux at 2000m, and organic matter burial at the sea floor (Kriest et al., 2017). In a subsequent optimisation Kriest (2017) kept the four optimal plankton parameters fixed, and calibrated four parameters related to remineralisation and nitrogen fixation (namely $K_{O2}$, $K_{DIN}$, $DIN_{min}$ and $\mu_{NFix}$ described in subsection 2.4), together with $b$ and $R_{-O2:P}$ (see Table 1). This second optimisation by Kriest (2017) led to a good match to independent estimates of pelagic denitrification, and is hereafter referred to as MIT28*. It serves as the starting point for the four additional optimisations presented in this paper."*

Line 347: "MOPS coupled to UVic circulation is more robust with respect to changes in parameters." : the sentence meaning is unclear, does this mean that the calibrated parameters are similar across the three UVic circulation used? (In comparison to ECCO or MIT28).

We will replace this by *"The misfit function changes less when the optimal parameters are swapped among the different UVic circulations."*

Line 349: ". . .the large impact of oxygen on the misfit function. . .": could you elaborate briefly why this is the case here.

We will rephrase this by *"In the model the global oxygen inventory adjusts dynamically to the combined effects of circulation and biogeochemical parameters, causing a large impact of this tracer on the misfit function (Kriest et al., 2017). Therefore, optimisation attempts to reduce the global oxygen bias, which is low for each optimal model configuration, indicated by the low values along the main diagonal of Figure 7, panel (B)."*

Lines 399-413: this analysis assumes that the interactions between circulation, biogeochemistry parameters and the misfit are linear and additive? Figure 11 suggests that this might not be the case as the delta_par and delta_circ bars do not sum to the delta_all bar. The analysis in this form is fine (and considering non-linear interactions would not be easy!) but I think this assumption should be mentioned.

Indeed, the fact that the individual contributions of delta_par and delta_circ do not add up to delta_all indicates that the effects are not linear and additive. Thank you for pointing this out. We will add a sentence on this at the end of this section: *"We note that the individual contributions of $\triangle Par$ and $\triangle Circ$ for both diagnostics do not add up to $\triangle All$ (Table 3), indicating that the effects of biogeochemical parameters and circulation are not linear and additive."*

Lines 453-454: ". . .it prevents fast settling of organic particles out the euphotic zone" is a little unclear. Does this mean there is effectively an increased residence time of particles in the euphotic zone which equates to a larger fraction of particles being remineralised before reaching the ocean interior? Is there also an impact of the plankton model in this instance, e.g., changes in zooplankton grazing?

Because the plankton parameters have not been changed during optimisation, global grazing follows almost linearly primary production (r=0.95), and all the statements and conclusions made with respect to the latter flux apply. In particular, a larger $b$ (slower settling; longer particle retention time at the surface) leads to an enhanced nutrient turnover in the euphotic zone; but as for primary production, this also depends on the circulation. We suggest to change this paragraph to make this clearer: *"(...) and thus primary production as the ultimate source of export production; on the other hand, it prevents fast settling of organic particles out of the euphotic zone. Because the plankton parameters were not changed during optimisation, global grazing follows almost linearly primary production (r=0.95), and the statements and conclusions made with respect to the former flux largely apply to grazing (no figure). Therefore, the combined antagonistic effects of $b$ on surface (and subsurface) nutrient turnover, subsurface nutrient concentrations (as a source of nutrient entrainment and mixing) and direct organic particle flux in the upper few hundred meters explain the relatively small variation caused by biogeochemical parameters (...)"*

Lines 459-461: "long term storage of nutrients and carbon will, to a large extent, depend on the prescribed particle flux profile"-the air-sea balance of CO2 might depend on circulation more than nutrients to the gas exchange component, similarly to the arguments made about O2 previously.

We agree, and suggest to rephrase this as *"Therefore, simulated organic matter supply to the deep ocean and deep nutrient concentrations will, to a large extent, depend on the prescribed particle flux profile, with potential effects on the long-term storage of carbon dioxide."*

Line 469-474: There should be a caveat that these findings are for MOPS specifically.

We agree, and will add: *"Therefore, at least for this particular biogeochemical model,..."*

Figure 4D is very hard to interpret due to the colour contrasts and placement/combination of lines. The panel is not explicitly mentioned in the text so I would

suggest to move the figure to supplementary or separate into more panels to make it clearer.

Because the percentage deviation especially in the deep ocean is just complementary, yet for some people important information, we now moved this panel to the supplement (as an additional plot), and have added a legend for the line colours and thicknesses, to make the plot more easily accessible.

---

## Author Comment (AC2) · 6 Apr 2020

We thank referee 2 very much for his/her encouraging and helpful detailed comments! Please find below our reply and answer to the referee's suggestions and comments (in blue).

Specific comments: Line 28: the use of two maxima here is a little confusing; can this be clarified at all?; a range of 180% might even be explicable by such mismatch of OMZ definitions, but I doubt that's what's happened here

Indeed, for some OMZ criteria the range reported by Bopp et al. (2013) is even larger than the 180% stated here, and very large (more than 20 times the observed volume) for a rather low OMZ criterion of 5 mmol O2/m3. We will replace this sentence

by: *"Bopp et al. (2013) report a range of variation that is several times the observed volume, depending on the criterion (maximum oxygen concentration) used for OMZ definition."*

Line 38: "by hand" is, in part, related to keeping the number of tuning simulations down; were a model to be inexpensive to run, automatic tuning sampling a large number of parameter sets would likely be preferred

We will add: *"To keep the number of computationally expensive, global simulations low, usually (...)"*

Line 59: "seems to be" -> "has proven to be"

Will be changed.

Lines 60-69: a very nice framing of the problem; thanks

Line 81: it seems remiss not to include even a sentence or two explaining what the TMM is (or what it comprises); it would spare your readers to present something here

We will add a few sentences on this: *"All model simulations and optimisations apply the Transport Matrix Method (Khatiwala, 2007, 2018), as an efficient "offline" method for ocean passive tracer transport. The TMM represents advection and mixing in the form of transport matrices that have been calculated from an ocean circulation model simulation prior to the biogeochemical simulations performed here. For our model simulations we apply monthly mean transport matrices (TMs), as well as monthly wind speed, temperature and salinity for air-sea gas exchange."*

Line 87: move this domain information into a table instead?

We would prefer to refrain from adding more information to table 1 (which gives the spatial resolution), and also rather not add another table to the main paper. However, we now cross-reference between this section and Table 1.

Line 104: interesting numbers are reported for this domain, but not the others; it would

be good to have all, or at least some other common information across the TMs

Because the original circulation models differ among each other (in their configuration), and because the offline circulation (in form of TMs) might differ from the basic circulation model we'd prefer not to extend too much on the original circulation model properties, but rather focus on the properties we can derive from the offline circulation. However, we suggest to present the physical diagnostics discussed in subsections 2.2 and 2.3 in an additional table in the supplement. (See also below.)

Lines 109-110: what does "accurately represent" here mean?; is it in reference to the circulation strengths you mention, or something else?

What we wanted to say in in fact, that these modified circulations have not been compared or evaluated agains the previously published circulations. We suggegstto replace this by *"We note that none of the circulation configurations, aside from UHigh, have been previously evaluated against the most commonly used UVic ESCM FCT configuration (e.g., Weaver et al., 2001; Schmittner et al., 2005; Somes et al., 2013)."*

Line 123: could a calculation of ventilation timescale (mean, max, and between basins) help separate out the differences between the TMs?

This is what we tried using the ideal age tracer. See below, we suggest to add the values of ideal age of different water masses, as well as globally to Table S1 in the supplement.

Line 129: "up to more than 800 years"; any idea why?; that does sound surprising at face value

One possible explanation of the strong increase in ideal age could be a lack of bottom water formation and ventilation in this model, which affects, for example CFC-11 (Dutay et al., 2002). In fact, the mixing criterion applied in our paper (>400m) would be too shallow to mix young surface and old deep waters, anyway. We therefore propose to skip *", despite its large area of deep mixing"*, and not give an explanation in this

subsection. Instead, we propose to mention it below, at the beginning of subsection 3.2 (see below, our reply to Line 169).

Line 130: ah-ha - ages mentioned here, but perhaps these could be added to the table I mentioned before for clarity

See above; we would rather not "burden" the table with this information, but have added a table on outcrop area, area of MLD and ideal age in the different water masses and globally to the supplement.

Line 132: "tracer observations"?; do you mean radiocarbon?

Yes, but also CFCs, temperature, salinity, phosphate, and oxygen. We will change this sentence to: *"constrained with transient (radiocarbon, CFCs) and hydrographic (temperature, salinity, nutrients, oxygen) tracer observations (Khatiwala et al., 2012)."*

Lines 137-138: this section feels like it could do with a sentence explaining how this information will be used later; however, it's certainly very helpful to elucidate how models might be good / bad

See our reply below, we have rewritten the first paragraph of this section and added a sentence on this.

Line 169: per my earlier remark, how's Drake Passage transport in the models?; it has a relationship with the SO properties mentioned

The transport through the Drake passage might, for example influence the properties and formation of SAMW in the Atlantic section (e.g., Sallee et al., 2010, 2013). However, as noted above we would prefer not to discuss details of the underlying circulation models too much in this paper. We propose to rewrite the beginning of subsection 2.3 as: *"The underlying circulation models, from which the TMs and forcing were extracted, differ in many aspects, such as parameterisation of mixing, forcing, sea-ice, etc., all of which can affect their dynamic behaviour and the quantities and diagnostics described above. For example, in the Southern Ocean the eastward transport of waters through*

*the Drake passage can affect the properties and formation of SAMW (e.g., Sallee et al., 2010, 2013), while parameterisation of sea ice in the models might affect the formation and ventilation of AABW (Dutay et al., 2002), with consequences for water mass age. It is beyond the scope of this paper to compare and discuss the details of the circulation models. Instead, to examine the potential impact of their charactistic features on optimal biogeochemical model parameters (section 3.2), we will focus on three diagnostics that can be easily derived from most circulation models (see Table S1 for simulated and observed values)."*

Line 204: how is near steady state defined? (presumably in Kriest et al., 2017 ...)

We have examined the transient behaviour of MOPS coupled to ECCO TMs over a simulation time of 9000 years in Kriest and Oschlies (2015). In that paper we focused on global average oxygen and nitrate (or fixed nitrogen), as these two properties are subject to processes affected by a variety of time scales (large scale circulation; deep remineralisation; air-sea gas exchange or fixed nitrogen gain and loss in different ocean basins). In the analysis we found that after significant variations within the first 3000 years (even with inflection points) the inventories begin to stabilise. We propose to add a note on this, with reference to that paper (in particular: Figure 2): *"Following the simulation of these 10 model setups over 3000 years to near-steady state, after which, for example, global oxygen and nitrate inventory change only by a small amount (see Figure 2 by Kriest and Oschlies, 2015), (...)"*

Line 210: "many local optima"; good - this is a perennial problem with BGC models

Yes, unfortunately.

Line 214: as these properties are tightly constrained in the real ocean (e.g. via the N:P ratio), is there an advantage to using all of them?; i.e. could N, O2 or P, O2 be sufficient?; something like carbon - which has a more plastic relationship with nutrients - might arguably be good too

In the model all three tracers - phosphate, nitrate, oxygen - are subject to different biogeochemical and physical processes, and this is the reason why we are using all of them in the combined misfit function: phosphate, as a conserved property, is not affected by a global bias (and therefore has the least impact on the misfit function; see Kriest et al., 2017). The oxygen inventory can adapt to the combined physical (air-sea gas exchange; subduction and ventilation) and biogeochemical (particle sinking and remineralisation) processes. So can nitrate, as its distribution in the ocean is affected by production, export, sinking and remineralisation. In addition, it is also affected by denitrification and nitrogen fixation, which happen in different areas, connected through large scale circulation as well as (local or regional) processes (e.g., upwelling). There-fore, we think that especially the latter two tracers (nitrate and oxygen) are necessary for the optimisation a model that simulates all three tracers independently. We agree, that at a later point it will be important to include other types of observations, such as DIC and alkalinity; but this would require a model that also parameterises the (ef-fects of) calcification, calcite sinking and dissolution. Finally, given the so far rather unconstrained biological model components (phyto- and zooplankton, detritus, DOP), we think that it might be even more important to include corresponding observations (even if they are sparse in space and time) into the optimisation.

Line 239: you could add the total number of simulations here to indicate the total computational load

We will add a line on this to table 1.

Line 245: earlier, in relation to burial and riverine input, you suggest that the inventory is not actually fixed, and can drift by a few percent; which is right?

Actually, we made a mistake here, and should have skipped "inventory" when referring to all nutrients at the end of section 2.4. Thank you for pointing this out. The form of resupply of buried P (and N) via either river runoff or surface supply alters the spatial distribution of nutrients and oxygen, as well as the inventory of nitrate and oxygen, but

the phosphorus inventory is fixed: global average phosphate remains almost the same (within 2e-4) between different model setups mentioned. (The very small change in phosphate inventory is likely due to faster equilibration when buried phosphate is distributed everywhere at the surface, and/or a slightly different contribution of organic P to total P, because of "fertilisation" of the surface.) Effects on nitrate and oxygen are larger, because these inventories can adapt dynamically to a different nutrient supply and turnover. This is currently subject to investigation. We will change the text accordingly: *"We note that in uncalibrated models (e.g., Kriest and Oschlies, 2015) this creates differences of a few percent in the global inventory of nitrate and oxygen. Differences in the regional distribution of nutrients and oxygen are largely comparable in magnitude to those caused by the numerical sinking scheme of detritus (Kriest and Oschlies, 2011). The effect of this process is subject to further research."*

Line 284: why especially oxygen?

As shown by Kriest et al. (2017), the misfit function is dominated by oxygen, because this tracer can include a global bias. The Pacific, with its very large volume, and old waters (which memorise the errors in biogeochemistry), is especially influential. We will add a reference to this paper and a comment: *"(...) and the comparatively large impact of oxygen on the misfit function (see Kriest et al., 2017)."*

Line 290: so that readers (like this one) don't have to scramble back a few pages, perhaps reiterate the default values once when you mention changes here

Kriest and Oschlies (2015) tested several values for the nitrate and oxygen affinity, so we find it difficult to refer to a default value here, but we suggest to mention the default values for max. rate of nitrogen fixation and the nitrate threshold of that paper, as well as that for $b$: *" (...) increasing b from a "default" value of 1.1 (Kriest and Oschlies, 2015) to 1.39. It further results in a high nitrate threshold for the onset of denitrification ($DIN_{\min} = 15.8$ mmol m$^{-3}$, compared to the default value of 4 mmol m$^{-3}$ applied by Kriest and Oschlies, 2015), a low affinity of denitrification to nitrate ($K_{DIN} = $*

$32$ *mmol m$^{-3}$), and a low maximum nitrogen fixation rate ($\mu_{\text{NFix}} = 1.19\,\mu$mol m$^{-3}$ d$^{-1}$; Table 1), which is only about half of the default value applied by Kriest and Oschlies (2015)."*

Line 362: it's not *completely* independent when the model is optimised to oxygen concentration; although, I appreciate it's not a target

We agree; we will add *"largely"* to this.

Line 428: should be "m3" not "m-3"; also, might want to contextualise with a percent of mean

Of course, thank you. We will add *"i.e., more than 100 the observed volume."* to put this into context.

Lines 639-640: "on the other hand . . ." is a confusing point; what do you mean?; it seems to suggest that a "benefit of parameter optimisation" is "helping to search for the best parameter set"; that sounds not particularly profound; something instead about "necessary level of model complexity"?

We agree, and suggest to rewrite this as: *"optimisation allows for a "fair" comparison of models of different complexity (after each model has been tuned to match some desired quantity best); it can therefore also help to decide about the necessary level of model complexity."* We will skip the part about model development.

Line 647: "through low ideal age"?

Changed.

Table 1: mean ocean ventilation age from these different circulations might be an interesting metric; or some other relevant integral metric of circulation

See above - we will add a table that provides the ideal ages in different water masses of all 5 circulations. Global average ideal age varies between 583 (UHigh) to 652 (MIT28), reflecting mainly the CDW, and will also given in supplementary table S1. However, as
we did not find a relation between any optimal parameter and the CDW age (Table 2), we prefer to not discuss this any further.

Table 3: export production - I like this illustrating of gaps in previous work (which has been more opportunistic than this study)

Figures: a general point I'd make is that red/blue colour bars are usually for situations where a property (e.g. a delta) has a definite central point worth marking (e.g. zero); here they're used broadly, potentially skewing the reader's perspective

We agree that red/blue might suggest a delta scale - on the other hand to our knowledge such a colour scale works well for visually impaired. We have now chosen a colour scale that does not bear such a strong resemblance to a delta scale, but should work for visually impaired as well (applies to Figures 1,2,3,5,S1,S2,S5) (see ferret.pmel.noaa.gov/static/FAQ/graphics/color_friendly_palettes.html)

Figure 1: you may have tried already, but might delta plots be better? (i.e. show the observations as field values but models as differences from this)

We tried, but to our opinion delta plot did not help interpretation.

Figure 2: as the observations are missing ice, I'd be inclined to skip it in the models as well

We will skip this.

Figure 3: see general and Figure 1 comments

See above, we changed to different colour scale.

Figure 4: panel D is rather complicated and ugly

We have skipped this panel, and added it separately as an additional figure in the supplement (see also comment by referee 1).

Figure 5: change to a much uglier palette; same red / blue issue

See above, we changed to different colour scale.

Figure 6: I really like this relating of parameter value to circulation property

Thank you. We have now added a symbol legend, based on a comment by reviewer 1.

Figures 8 and 9: I found these rather difficult to interpret, although I have no suggestions on how to change them

We are aware of the fact that these are difficult to interpret, and tried many different approaches, but also have not found any better was to illustrate this.

---

## Author Response (AR1)

Dear Editor and Referees,

please find below our detailed resposens to comments and suggestions by referees 1 and 2 (in blue). In general, they match our answers given in the public discussion. Three slight divergences the changes suggested in our the open discussion response to referee 1 are highlighted in bold.

**1 Reply to referee 1:**

General Comments: Some of the figures were a little hard to interpret due to the large number of model set-ups. Figures 6, 8, 9, 10 and 12 would really benefit from a legend or key to more quickly pick out which model set-up is which (a legend on the first relevant plot that is referenced for subsequent plots for example). Interpreting the plots with reference to the text was difficult because of this.

We have now added a symbol legend to Figure 6 and refer to this in plots 8, 9, 10 and 12. Note that we have also increased the number of digits in the regression equations (formerly truncated to 2-4).

Specific Comments: Section 2.7: I found this section a little unclear due to the discussion about the previous optimisations. It would help to focus on the parameters varied in this study based on the second 2017 optimisation and mention more briefly that the plankton parameters were the same as the first 2017 optimisation.

We found it somehow difficult to rephrase this by just mentioning the first optimisation by Kriest et al. (2017) in passing (and secondly), We have rewritten this subsection to hopefully make it easier to understand the sequence of optimisations: "The optimisations presented here are based upon two successive optimisations presented by Kriest et al. (2017) and Kriest (2017). Both studies applied model MOPS coupled to TMs derived from MIT28. The cost function, as presented in Equation 1 was calculated after a spin up of 3000 years, as also used in the present study. In the first optimisation, Kriest et al. (2017) optimised four parameters related to plankton growth and loss terms, together with b and  $R_{-O2:P}$ . The optimal parameters of this first calibration led to a better agreement of simulated global biogeochemical fluxes to observations of primary and export production, zooplankton grazing, particle flux at 2000 m, and organic matter burial at the sea floor (Kriest et al., 2017). In a subsequent optimisation Kriest (2017) kept the four optimal plankton parameters fixed, and calibrated four parameters related to remineralisation and nitrogen fixation (namely  $K_{O2}$ ,  $K_{DIN}$ ,  $DIN_{min}$  and  $\mu_{\rm NFix}$  described in section 2.4), together with b and  $R_{-\rm O2:P}$  (see Table 1). This second optimisation by Kriest (2017) led to a better match to independent estimates of pelagic denitrification, without deteriorating the matches to observed primary and export production, zooplankton grazing, particle flux at 2000m and organic matter burial at the sea floor. It is hereafter referred to as  $MIT28^*$  and serves as the starting point for the four additional optimisations presented in this paper. In particular, we repeated optimisation  $MIT28^*$  against (...)"

Line 347: "MOPS coupled to UVic circulation is more robust with respect to changes in parameters." : the sentence meaning is unclear, does this mean that the calibrated parameters are similar across the three UVic circulation used? (In comparison to ECCO or MIT28).

We have replaced this by "The misfit function changes less when the optimal parameters are swapped among the different UVic circulations."

Line 349: '. . . the large impact of oxygen on the misfit function. . .": could you elaborate briefly why this is the case here.

We have rephrased this by "In the model the global oxygen inventory adjusts dynamically to the combined effects of circulation and biogeochemical parameters, causing a large impact of this tracer on the misfit function (Kriest et al., 2017). Therefore, optimisation attempts to reduce the global oxygen bias, which is low for each optimal model configuration, indicated by the low values along the main diagonal of Figure 7, panel (B)."

Lines 399-413: this analysis assumes that the interactions between circulation, biogeochemistry parameters and the misfit are linear and additive? Figure 11 suggests that this might not be the case as the delta\_par and delta\_circ bars do not sum to the delta\_all bar. The analysis in this form is fine (and considering non-linear interactions would not be easy!) but I think this assumption should be mentioned.

We have changed this to "We note that the individual contributions of  $\Delta Par$  and  $\Delta Circ$  for both diagnostics do not add up to  $\Delta All$  (Table 3), indicating that the effects of biogeochemical parameters and circulation are not linear and additive."

Lines 453-454: ". . .it prevents fast settling of organic particles out the euphotic zone" is a little unclear. Does this mean there is effectively an increased residence time of particles in the euphotic zone which equates to a larger fraction of particles being rem- ineralised before reaching the ocean interior? Is there also an impact of the plankton model in this instance, e.g., changes in zooplankton grazing?

Because the plankton parameters have not been changed during optimisation, global grazing follows almost linearly primary production (r=0.95), and all the statements and conclusions made with respect to the latter flux apply. In particular, a larger b (slower settling; longer particle retention time at the surface) leads to an enhanced nutrient turnover in the euphotic zone; but as for primary production, this also depends on the circulation. We have changed this paragraph to make this clearer: "(...) and thus primary production as the ultimate source of export production; on the other hand, it prevents fast settling of organic particles out of the euphotic zone. Because the plankton parameters were not changed during optimisation, global grazing follows almost linearly primary production (r = 0.95), and the statements and conclusions made with respect to the former flux largely apply to grazing (no figure). Therefore, the combined antagonistic effects of b on surface (and subsurface) nutrient turnover, subsurface nutrient concentrations (as a source of nutrient entrainment and mixing) and direct organic particle flux in the upper few hundred meters explain the relatively small variation caused by biogeochemical parameters (...)"

Lines 459-461: "long term storage of nutrients and carbon will, to a large extent, depend on the prescribed particle flux profile"-the air-sea balance of CO2 might depend on circulation more than nutrients to the gas exchange component, similarly to the arguments made about O2 previously.

We agree, and have rephrased this as "Therefore, simulated organic matter supply to the deep ocean and deep nutrient concentrations will, to a large extent, depend on the prescribed particle flux profile, with potential effects on the long-term storage of carbon dioxide."

Line 469-474: There should be a caveat that these findings are for MOPS specifically. We have added: "(...) at least for this particular biogeochemical model, (...)"

Figure 4D is very hard to interpret due to the colour contrasts and placement/combination of lines. The panel is not explicitly mentioned in the text so I would suggest to move the figure to supplementary or separate into more panels to make it clearer.

We now moved this panel to the supplement (as an additional plot), and have added a legend for the line colours and thicknesses, to make the plot more easily accessible.

**2 Reply to referee 2:**

Specific comments: Line 28: the use of two maxima here is a little confusing; can this be clarified at all?; a range of 180% might even be explicable by such mismatch of OMZ definitions, but I doubt that's what's happened here

We have replaced this by: "(...) Bopp et al. (2013) report a range of variation that is several times the observed volume, depending on the criterion (maximum oxygen concentration) used for OMZ definition."

Line 38: "by hand" is, in part, related to keeping the number of tuning simulations down; were a model to be inexpensive to run, automatic tuning sampling a large number of parameter sets would likely be preferred We have added: "To keep the number of computationally expensive, global simulations low, usually (...)"

Line 59: "seems to be" -i "has proven to be" Changed.

Lines 60-69: a very nice framing of the problem Thank you!

Line 81: it seems remiss not to include even a sentence or two explaining what the TMM is (or what it comprises); it would spare your readers to present something here

We added a few sentences on this: "All model simulations and optimisations apply the Transport Matrix Method (Khatiwala, 2007, 2018), as an efficient "offline" method for ocean passive tracer transport. The TMM represents advection and mixing in the form of transport matrices that have been calculated from an ocean circulation model simulation prior to the biogeochemical simulations performed here. For our model simulations we apply monthly mean transport matrices (TMs), as well as monthly wind speed, temperature and salinity for air-sea gas exchange."

**Line 87: move this domain information into a table instead?**

We would prefer to refrain from adding more information to table 1 (which gives the spatial resolution), and also rather not add another table to the main paper. However, we now cross-reference between this section and

**Table 1.**

Line 104: interesting numbers are reported for this domain, but not the others; it would be good to have all, or at least some other common information across the TMs

Because the original circulation models differ among each other (in their configuration), and because the offline circulation (in form of TMs) might differ from the basic circulation model we'd prefer not to extend too much on the original circulation model properties, but rather focus on the properties we can derive from the offline circulation. However, we now present the physical diagnostics discussed in subsections 2.2 and 2.3 in an additional table in the supplement. (See also below.)

Lines 109-110: what does "accurately represent" here mean?; is it in reference to the circulation strengths you mention, or something else?

What we wanted to say in in fact, that these modified circulations have not been compared or evaluated agains the previously published circulations. We replaced this by "We note that none of the circulation configurations, aside from UHigh, have been previously evaluated against the most commonly used UVic ESCM FCT configuration (e.g., Weaver et al., 2001; Schmittner et al., 2005; Somes et al., 2013)."

**Line 123: could a calculation of ventilation timescale (mean, max, and between basins) help separate out the differences between the TMs?**

This is what we tried using the ideal age tracer. See below, we added the values of ideal age of different water masses, as well as globally to Table S1 in the supplement.

**Line 129: "up to more than 800 years"; any idea why?; that does sound surprising at face value**

One possible explanation of the strong increase in ideal age could be a lack of bottom water formation and ventilation in this model, which affects, for example CFC-11 (Dutay et al., 2002). In fact, the mixing criterion applied in our paper (>400m) would be too shallow to mix young surface and old deep waters, anyway. We therefore skipped ", despite its large area of deep mixing", but mention it below, at the beginning of subsection 3.2 (see below, our reply to Line 169).

**Line 130: ah-ha - ages mentioned here, but perhaps these could be added to the table I mentioned before for clarity**

See above; we would rather not "burden" the table with this information, but have added a table on outcrop area, area of MLD and ideal age in the different water masses and globally to the supplement.

**Line 132: "tracer observations"?; do you mean radiocarbon?**

Yes, but also CFCs, temperature, salinity, phosphate, and oxygen. We have changed this sentence to: "constrained with transient (radiocarbon, CFCs) and hydrographic (temperature, salinity, nutrients, oxygen) tracer observations (Khatiwala et al., 2012)."

Lines 137-138: this section feels like it could do with a sentence explaining how this information will be used later; however, it's certainly very helpful to elucidate how models might be good / bad

See our reply below, we have rewritten the first paragraph of this section and added a sentence on this.

**Line 169: per my earlier remark, how's Drake Passage transport in the models?; it has a relationship with the SO properties mentioned**

The transport through the Drake passage might, for example influence the properties and formation of SAMW in the Atlantic section (e.g., Sallee et al., 2010, 2013). However, as noted above we would prefer not to discuss details of the underlying circulation models too much in this paper. We have rewritten the beginning of subsection 2.3 as: "The underlying circulation models, from which the TMs and forcing were extracted, differ in many aspects, such as parameterisation of mixing, forcing, sea-ice, etc., all of which can affect their dynamic behaviour and the quantities and diagnostics described above. For example, in the Southern Ocean the eastward transport of waters through the Drake passage can affect the properties and formation of SAMW (e.g., Sallee et al., 2010, 2013), while parameterisation of sea ice in the models might affect the formation and ventilation of AABW (Dutay et al., 2002), with consequences for water mass age. It is beyond the scope of this paper to compare and discuss the details of the circulation models. Instead, to examine the potential impact of their charactistic features on optimal biogeochemical model parameters (section 3.2), we will focus on three diagnostics that can be easily derived from most circulation models (see Table S1 for simulated and observed values)."

Line 204: how is near steady state defined? (presumably in Kriest et al., 2017 ...)

We have examined the transient behaviour of MOPS coupled to ECCO TMs over a simulation time of 9000 years in Kriest and Oschlies (2015). In that paper we focused on global average oxygen and nitrate (or fixed

nitrogen), as these two properties are subject to processes affected by a variety of time scales (large scale circulation; deep remineralisation; air-sea gas exchange or fixed nitrogen gain and loss in different ocean basins). In the analysis we found that after significant variations within the first 3000 years (even with inflection points) the inventories begin to stabilise. We have now added a note on this, with reference to that paper (in particular: Figure 2): *"Following the simulation of these 10 model setups over 3000 years to near-steady state, after which,* for example, global oxygen and nitrate inventory change only by a small amount (see Figure 2 by Kriest and Oschlies, 2015), (...)"

Line 210: "many local optima"; good - this is a perennial problem with BGC models Yes, unfortunately.

Line 214: as these properties are tightly constrained in the real ocean (e.g. via the N:P ratio), is there an advantage to using all of them?; i.e. could N, O2 or P, O2 be sufficient?; something like carbon - which has a more plastic relationship with nutrients - might arguably be good too

In the model all three tracers - phosphate, nitrate, oxygen - are subject to different biogeochemical and physical processes, and this is the reason why we are using all of them in the combined misfit function: phosphate, as a conserved property, is not affected by a global bias (and therefore has the least impact on the misfit function; see Kriest et al., 2017). The oxygen inventory can adapt to the combined physical (air-sea gas exchange; subduction and ventilation) and biogeochemical (particle sinking and remineralisation) processes. So can nitrate, as its distribution in the ocean is affected by production, export, sinking and remineralisation. In addition, it is also affected by denitrification and nitrogen fixation, which happen in different areas, connected through large scale circulation as well as (local or regional) processes (e.g., upwelling). Therefore, we think that especially the latter two tracers (nitrate and oxygen) are necessary for the optimisation a model that simulates all three tracers independently. We agree, that at a later point it will be important to include other types of observations, such as DIC and alkalinity; but this would require a model that also parameterises the (effects of) calcification, calcite sinking and dissolution. Finally, given the so far rather unconstrained biological model components (phytoand zooplankton, detritus, DOP), we think that it might be even more important to include corresponding observations (even if they are sparse in space and time) into the optimisation.

Line 239: you could add the total number of simulations here to indicate the total computational load We added a line on this to table 1.

**Line 245: earlier, in relation to burial and riverine input, you suggest that the inventory is not actually fixed, and can drift by a few percent; which is right?**

Actually, we made a mistake here, and should have skipped "inventory" when referring to all nutrients at the end of section 2.4. Thank you for pointing this out. The form of resupply of buried P (and N) via either river runoff or surface supply alters the spatial distribution of nutrients and oxygen, as well as the inventory of nitrate and oxygen, but the phosphorus inventory is fixed: global average phosphate remains almost the same (within 2e-4) between different model setups mentioned. (The very small change in phosphate inventory is likely due to faster equilibration when buried phosphate is distributed everywhere at the surface, and/or a slightly different contribution of organic P to total P, because of "fertilisation" of the surface.) Effects on nitrate and oxygen are larger, because these inventories can adapt dynamically to a different nutrient supply and turnover. This is currently subject to investigation. We have changed the text at the end of section 2.4 accordingly: "We note that in uncalibrated models (e.g., Kriest and Oschlies, 2015) this creates differences of a few percent in the global inventory of nitrate and oxygen. Differences in the regional distribution of nutrients and oxygen are largely comparable in magnitude to those caused by the numerical sinking scheme of detritus (Kriest and Oschlies, 2011). The effect of this process is subject to further research."

**Line 284: why especially oxygen?**

As shown by Kriest et al. (2017), the misfit function is dominated by oxygen, because this tracer can include a global bias. The Pacific, with its very large volume, and old waters (which memorise the errors in biogeochemistry), is especially influential. We added a reference to this paper and a comment: "(...) and the comparatively large impact of oxygen on the misfit function (see Kriest et al., 2017)."

Line 290: so that readers (like this one) don't have to scramble back a few pages, perhaps reiterate the default values once when you mention changes here

Kriest and Oschlies (2015) tested several values for the nitrate and oxygen affinity, so we find it difficult to refer to a default value here, but we now mention the default values for max. rate of nitrogen fixation and the nitrate threshold of that paper, as well as that for b: "(...) increasing b from a "default" value of 1.1 (Kriest and Oschlies, 2015) to 1.39. It further results in a high nitrate threshold for the onset of denitrification  $(DIN_{\min} = 15.8 \text{ mmol } m^{-3}, \text{ compared to the default value of 4 mmol } m^{-3} \text{ applied by Kriest and Oschlies},$

2015), a low affinity of denitrification to nitrate ( $K_{\text{DIN}} = 32 \text{ mmol } m^{-3}$ ), and a low maximum nitrogen fixation rate ( $\mu_{\text{NFix}} = 1.19 \,\mu\text{mol } m^{-3} \, d^{-1}$ ; Table 1), which is only about half of the default value applied by Kriest and Oschlies (2015)."

Line 362: it's not \*completely\* independent when the model is optimised to oxygen concentration; although, I appreciate it's not a target We added *"largely"*.

Line 428: should be "m3" not "m-3"; also, might want to contextualise with a percent of mean Corrected, thank you. We added *"i.e., more than 100 the observed volume."* to put this into context.

Lines 639-640: "on the other hand . . ." is a confusing point; what do you mean?; it seems to suggest that a "benefit of parameter optimisation" is "helping to search for the best parameter set"; that sounds not particularly profound; something instead about "necessary level of model complexity"?

We agree, and rewrote this as: "optimisation allows for a "fair" comparison of models of different complexity (after each model has been tuned to match some desired quantity best); it can therefore also help to decide about the necessary level of model complexity." We skipped the part about model development.

Line 647: "through low ideal age"? Changed.

Table 1: mean ocean ventilation age from these different circulations might be an interesting metric; or some other relevant integral metric of circulation

See our comments above. We now added a table that provides the ideal ages in different water masses of all 5 circulations. Global average ideal age varies between 583 (UHigh) to 652 (MIT28), reflecting mainly the CDW, and is now also given in supplementary table S1. However, as we did not find a relation between any optimal parameter and the CDW age (Table 2), we do not discuss this any further.

Table 3: export production - I like this illustrating of gaps in previous work (which has been more opportunistic than this study) Thank you!

Figures: a general point I'd make is that red/blue colour bars are usually for situations where a property (e.g. a delta) has a definite central point worth marking (e.g. zero); here they're used broadly, potentially skewing the reader's perspective

We agree that red/blue might suggest a delta scale - on the other hand to our knowledge such a colour scale works well for visually impaired. We have now chosen a colour scale that does not bear such a strong resemblance to a delta scale, but should work for visually impaired as well (applies to Figures 1,2,3,5,S1,S2,S5) (see ferret.pmel.noaa.gov/static/FAQ/graphics/color\_friendly\_palettes.html)

Figure 1: you may have tried already, but might delta plots be better? (i.e. show the observations as field values but models as differences from this)

We tried, but to our opinion delta plot did not help interpretation.

Figure 2: as the observations are missing ice, I'd be inclined to skip it in the models as well We skipped the hatched area for sea ice.

Figure 3: see general and Figure 1 comments See above, we changed to different colour scale.

Figure 4: panel D is rather complicated and ugly

We have skipped this panel, and added it separately as an additional figure in the supplement (see also comment by referee 1).

Figure 5: change to a much uglier palette; same red / blue issue See above, we changed to different colour scale.

Figure 6: I really like this relating of parameter value to circulation property

Thank you. We have now added a symbol legend, based on a comment by reviewer 1.

Figures 8 and 9: I found these rather difficult to interpret, although I have no suggestions on how to change them

We are aware of the fact that these are difficult to interpret, and tried many different approaches, but also have not found any better was to illustrate this.

[revised manuscript text omitted]
 optimisatic           | on perfor | mance:                        |       |                               |       |                               |       |                               |       |                               |
| $\widetilde{N}$                          |           | 119                           |       | 144                           |       | 125                           |       | 130                           |       | $100^{\dagger}$               |
| $J^*$                                    |           | 0.439                         | -     | 0.401                         | -     | 0.431                         | -     | 0.489                         | _     | 0.366                         |
| Bias $NO_3$ (mmol m -3 )      |           | -0.27                         |       | 0.68                          |       | 0.30                          |       | 0.52                          |       | 0.28                          |
| Bias $O_2 \pmod{m^{-3}}$                 |           | 3.03                          |       | 1.63                          |       | 3.56                          |       | 5.29                          |       | 0.08                          |
| Bias OMZ Vol. 50 $(10^{15} \text{ m}^3)$ |           | -19.0                         |       | -5.7                          |       | 3.1                           |       | 12.6                          |       | -23.2                         |
| Bias OMZ Vol. 80 $(10^{15} \text{ m}^3)$ |           | -44.0                         |       | 16.8                          |       | 11.7                          |       | 22.7                          |       | -40.7                         |
| Parameters [Range]                       |           |                               |       |                               |       |                               |       |                               |       |                               |
| b                                        | 1.39      | [1.3-1.5]                     | 1.27  | [1.2 - 1.3]                   | 1.40  | [1.3-1.5]                     | 1.44  | [1.3-1.6]                     | 1.46  | [1.4-1.5]                     |
| $R_{-02:\mathrm{P}}$                     | 173.7     | [166-178]                     | 161.5 | [158-166]                     | 161.5 | [158-167]                     | 167.0 | [159-175]                     | 151.1 | [150-154]                     |
| $\mu_{\rm NFix}$                         | 1.19      | [1.1-1.8]                     | 2.98  | [2.1-3.0]                     | 1.00  | [1.0-1.4]                     | 1.98  | [1.5-3.0]                     | 2.29  | [1.5-3.0]                     |
| $DIN_{ m min}$                           | 15.8      | [12.2-16.0]                   | 16.0  | [6.2 - 16.0]                  | 15.9  | [7.0-16.0]                    | 16.0  | [12.0-16.0]                   | 16.0  | [10.2-16.0]                   |
| $K_{ m DIN}$                             | 32.0      | [17.2-32.0]                   | 26.5  | [17.4 - 31.4]                 | 31.9  | [25.8-32.0]                   | 32.0  | [21.3-32.0]                   | 23.1  | [21.7-32.0]                   |
| $K_{02}$                                 | 1.01      | [1.0-8.5]                     | 13.58 | [2.1-14.3]                    | 7.15  | [1.0-8.0]                     | 5.62  | [1.5-16.0]                    | 1.07  | [1.0-16.0]                    |

**Table 2.** Correlation coefficient *r* for the regression of three optimal model parameters against physical diagnostics. The outcrop area of dense waters in the northern (> 40°N) and southern (> 40°S) hemisphere is given for two different density intervals. Outcrop area for deep mixing in the North Atlantic (> 40°N) and Southern Ocean (> 40°S) is given for two different criteria of maximum deep mixing (200m and 400m). Mean water mass age is given for four different water masses. See text for further details. Significant correlations (p < 0.05) are denoted by bold face.

| Parameter            | Physical diagnostic     |              |                        |            |  |  |  |
|----------------------|-------------------------|--------------|------------------------|------------|--|--|--|
|                      | Area MLD North Atlantic |              | Area MLD Southern Ocea |            |  |  |  |
|                      | 200m                    | 400m         | 200m                   | 400m       |  |  |  |
| $R_{-\mathrm{O2:P}}$ | 0.916                   | 0.950        | 0.875                  | 0.465      |  |  |  |
| $\mu_{ m NFix}$      | -0.492                  | -0.405       | -0.332                 | -0.263     |  |  |  |
| b                    | -0.280                  | -0.425       | -0.499                 | -0.210     |  |  |  |
|                      | Area Outo               | crop North   | Area Out               | crop South |  |  |  |
|                      | 26.5-27.5               | >27.5        | 26.5-27.5              | >27.5      |  |  |  |
| $R_{-\mathrm{O2:P}}$ | -0.329                  | 0.460        | -0.641                 | 0.792      |  |  |  |
| $\mu_{ m NFix}$      | 0.449                   | -0.361       | 0.890                  | -0.746     |  |  |  |
| b                    | -0.287                  | -0.287 0.050 |                        | 0.202      |  |  |  |
|                      |                         | Idea         | l Ages                 |            |  |  |  |
|                      | NADW CDW                |              | NPDW                   | ETP        |  |  |  |
| $R_{-O2:P}$          | -0.358                  | 0.273        | 0.787                  | -0.266     |  |  |  |
| $\mu_{\rm NFix}$     | -0.083                  | -0.380       | -0.857                 | -0.070     |  |  |  |
| b                    | 0.914 0.136      |              | 0.069                  | 0.334      |  |  |  |

**Table 3.** Mean (across all experiments and optimal models) and range of variation of global biogeochemical model properties and fluxes across different parameter sets (circulation constant;  $\Delta$  Par), and different circulations (parameters constant;  $\Delta$  Circ), as well as across the five different optimal models MIT28\*, ECCO\*, uHigh\*, U20\* and U17.5\* ( $\Delta$  Opt).  $\Delta$  Mod shows the difference between MIT28\* of this study and model RetroMOPS of Kriest (2017). Observed oxygen and OMZ volume from Garcia et al. (2006b, mapped onto ECCO grid); observed global flux ranges derived from estimates by Carr et al. (2006, primary production), Dunne et al. (2007, export production), Lutz et al. (2007, export production; radiogen. calib.), Honjo et al. (2008, mean particle flux), Guidi et al. (2015, particle flux), Eugster and Gruber (2012, median fixed nitrogen loss) and Somes et al. (2013, fixed nitrogen loss of best data-constrained model).

|                                   | Mean (All)      | Mean (Opt)                | $\Delta \operatorname{Mod}$ | $\Delta$ Par         | $\Delta{\rm Circ}$            | $\Delta$ Opt | $\Delta$ All |
|-----------------------------------|-----------------|---------------------------|-----------------------------|----------------------|-------------------------------|--------------|--------------|
|                                   | Global mean     | O 2 (observed: | 174.17 mn                   | nol $m^{-3}$ )       |                               |              |              |
| This study                        | 177.2           | 177.6                     | 0.1                         | 24.1                 | 24.3                          | 6.8          | 43.0         |
| Bopp et al. (2013)                |                 |                           |                             |                      |                               |              | 95.0         |
| Ol                                | MZ Volume (50   | mmol m $^{-3}$ ; ot       | oserved: 57                 | $0.0 \times 10^{15}$ | m 3 )              |              |              |
| This study                        | 54.9            | 52.1                      | 0.7                         | 39.4                 | 55.4                          | 39.0         | 67.2         |
| Bopp et al. (2013)                |                 |                           |                             |                      |                               |              | 212.5        |
| ON                                | AZ Volume (80   | mmol $m^{-3}$ ; ob        | served: 119                 | $9.1 \times 10^{13}$ | 5 m 3 ) |              |              |
| This study                        | 122.3           | 112.8                     | 0.6                         | 112.0                | 119.8                         | 73.3         | 145.9        |
| Bopp et al. (2013)                |                 |                           |                             |                      |                               |              | 328.9        |
|                                   | Fixed N loss (  | observed estima           | ates: 52-76                 | $Tg N y^{-1}$        | L )                |              |              |
| This study                        | 67.7            | 67.4                      | 1.9                         | 35.1                 | 33.8                          | 17.4         | 44.9         |
| Somes et al. (2013) \$ |                 |                           | 53.6                        |                      |                               |              |              |
| Pr                                | imary Productio | on (observed est          | timates: 40                 | -60 Pg C             | y -1 )             |              |              |
| This study                        | 47.4            | 47.2                      | 9.1                         | 8.17                 | 12.27                         | 5.65         | 18.95        |
| Schmittner et al. $(2005)^{\S}$   |                 |                           |                             | 20.1                 | 24.5                          |              | 48.5         |
| Seferian et al. (2013)            |                 |                           |                             |                      | 8.64                          |              |              |
| Bopp et al. (2013)                |                 |                           |                             |                      |                               |              | 47.8         |
| Ex                                | port Production | n (observed esti          | mates: 4.6-                 | 9.6 Pg C             | y -1 )             |              |              |
| This study                        | 6.86            | 6.86                      | 0.23                        | 0.49                 | 1.08                          | 1.05         | 1.20         |
| Schmittner et al. $(2005)^{\S}$   |                 |                           |                             | 2.2                  | 8.4                           |              | 11.2         |
| Najjar et al. (2007)              |                 |                           |                             |                      | 10                            |              |              |
| Kwon et al. $(2009)^{\dagger}$    |                 |                           |                             | $\approx 5$          |                               |              |              |
| Seferian et al. (2013)            |                 |                           |                             |                      | 3.0                           |              |              |
| Bopp et al. (2013)                |                 |                           |                             |                      |                               |              | 3.2          |
| Sedim                             | entation (2000r | n) (observed es           | timates: 0.3                | 33-0.43 P            | $g C y^{-1}$ )                |              |              |
| This study                        | 0.36            | 0.36                      | 0.02                        | 0.16                 | 0.06                          | 0.16         | 0.19         |
| Najjar et al. $(2007)^{\ddagger}$ |                 |                           |                             |                      | 0.52                          |              |              |

<sup>§ experiments 1-3; § For  $\Delta$ Circ we have omitted the experiment with  $K_b = 0$  of the study by Schmittner et al. (2005), and refer only to experiments 2, 8 and 12. For  $\Delta$ Par we report the difference between experiments 12 and 13, and for  $\Delta$ All the maximum spread across experiments 2 to 13. † We refer to Figure 2b of Kwon et al. (2009), but consider only a range of  $1.1 \le b \le 1.4$  for  $\Delta$ Par, to be comparable to our range of *b*. ‡ Calculated from export production  $\times (75/2000)^{0.9} = 0.052$ .